# Optogenetic manipulation of neuronal and cardiomyocyte functions in zebrafish using microbial rhodopsins and adenylyl cyclases

Hanako Hagio[1,2,3†], Wataru Koyama[1†], Shiori Hosaka[1], Aysenur Deniz Song[1], Janchiv Narantsatsral[1], Koji Matsuda[1], Takashi Shimizu[1], Shoko Hososhima[4], Satoshi P Tsunoda[4], Hideki Kandori[4], Masahiko Hibi[1]*

[1]Graduate School of Science, Nagoya University, Japan, Nagoya, Japan; [2]Graduate School of Bioagricultural Sciences, Nagoya University, Nagoya, Japan; [3]Institute for Advanced Research, Nagoya University, Nagoya, Japan; [4]Department of Life Science and Applied Chemistry, Nagoya Institute of Technology, Nagoya, Japan

*For correspondence:
hibi.masahiko.s7@f.mail.nagoya-u.ac.jp

†These authors contributed equally to this work

Competing interest: The authors declare that no competing interests exist.

## Abstract

Even though microbial photosensitive proteins have been used for optogenetics, their use should be optimized to precisely control cell and tissue functions in vivo. We exploited *Gt*CCR4 and *Kn*ChR, cation channelrhodopsins from algae, *Be*GC1, a guanylyl cyclase rhodopsin from a fungus, and photoactivated adenylyl cyclases (PACs) from cyanobacteria (*Oa*PAC) or bacteria (*b*PAC), to control cell functions in zebrafish. Optical activation of *Gt*CCR4 and *Kn*ChR in the hindbrain reticulospinal V2a neurons, which are involved in locomotion, induced swimming behavior at relatively short latencies, whereas activation of *Be*GC1 or PACs achieved it at long latencies. Activation of *Gt*CCR4 and *Kn*ChR in cardiomyocytes induced cardiac arrest, whereas activation of *b*PAC gradually induced bradycardia. *Kn*ChR activation led to an increase in intracellular $Ca^{2+}$ in the heart, suggesting that depolarization caused cardiac arrest. These data suggest that these optogenetic tools can be used to reveal the function and regulation of zebrafish neurons and cardiomyocytes.

## Editor's evaluation

This manuscript provides a valuable resource for scientists who wish to manipulate second messengers in zebrafish using optogenetics. The authors provide solid evidence, based on behaviour, monitoring of heartbeat and imaging, that several of the tools tested can have an effect in larval fish. Tools that lack an effect are also described. As the tools affect second messengers that are used in multiple cell types, the results should be of interest to scientists working in a variety of areas.

## Introduction

Cells can respond to various signals by changing their internal states. For example, in the nervous system, neurons respond to neurotransmitters to increase or decrease ions and/or chemical mediators such as cAMP and cGMP in the cytoplasm. Similarly, cardiomyocyte function is regulated by sympathetic and parasympathetic nerves that involve noradrenergic and cholinergic receptors, respectively, and control chemical mediators such as cAMP. To understand the regulation of cell and tissue functions, it is necessary to manipulate intracellular ions and cAMP/cGMP at a precise timing and locations, and examine their effects on cell and tissue functions in vivo.

Optogenetics is a rapidly expanding technology that controls or detects cellular functions by using photoreactive proteins that are genetically expressed in cells. Microbial rhodopsins have been used for optogenetics (*Kandori, 2020*; *Kandori, 2021*). Two main types of microbial rhodopsins are used in optogenetics. The first includes microbial rhodopsins with ion-transporting properties such as light-gated ion channels (channelrhodopsins [ChRs]) and light-driven ion pumps, while the other includes microbial rhodopsins with enzymatic activity, that is, enzymorhodopsins (Figure 2B).

Among the ChRs, channelrhodopsin 1 and 2 (*Cr*ChR1 and *Cr*ChR2), which are light-gated cation channels, were identified from the green alga *Chlamydomonas reinhardtii* (*Nagel et al., 2002*; *Sine-shchekov et al., 2002*; *Suzuki et al., 2003*). When *Cr*ChR1 and *Cr*ChR2 were expressed in *Xenopus* oocytes, they functioned as light-gated cation-selective channels (*Nagel et al., 2002*; *Nagel et al., 2003*). *Cr*ChR2 was used to depolarize mammalian cells in response to light (*Boyden et al., 2005*; *Ishizuka et al., 2006*). Thereafter, variants of *Cr*ChR2 or chimeric forms of *Cr*ChR1 and *Cr*ChR2 were developed to improve the efficiency of expression and induce higher activity than the original *Cr*ChR1 and *Cr*ChR2 (*Berndt et al., 2011*; *Deisseroth, 2011*; *Ernst et al., 2014*; *Wang et al., 2009*). These efforts have made *Cr*ChRs the most commonly used optogenetic tools to induce depolarization in neurons and mimic neuronal activation through ion channel-type neurotransmitter receptors. However, there are some limitations when using *Cr*ChRs as optogenetic tools. The ion selectivity of *Cr*ChR2 is much higher for $H^+$ than $Na^+$ (*Nagel et al., 2003*). If pH inside and outside the cell differs, *Cr*ChR2 acts as an $H^+$ channel rather than an $Na^+$ channel. As *Cr*ChR2 is permeable to $Ca^{2+}$ to some extent (*Nagel et al., 2003*), neuronal activation by *Cr*ChR2 leads to both depolarization and activation of the $Ca^{2+}$ pathway, making it difficult to distinguish between the two effects. *Gt*CCR4 is a light-gated cation channel derived from the cryptophyte *Guillardia theta* (*Govorunova et al., 2016*; *Yamauchi et al., 2017*). The light sensitivity of *Gt*CCR4 is higher than that of *Cr*ChR2 while the channel open lifetime lies in the same range as that of *Cr*ChR2 when expressed in mammalian neuronal cells. Since *Gt*CCR4 conducts almost no $H^+$ and no $Ca^{2+}$ under physiological conditions, *Gt*CCR4 is a high $Na^+$-selective ChR (*Hososhima et al., 2020*; *Shigemura et al., 2019*). *Kn*ChR is another cation ChR derived from the filamentous terrestrial alga *Klebsormidium nitens* (*Tashiro et al., 2021*). Truncation of the carboxy-terminal of *Kn*ChR prolonged the channel open lifetime by more than 10-fold, providing strong light-induced channel activity (*Tashiro et al., 2021*). These findings imply that *Gt*CCR4 and truncated variants of *Kn*ChR are alternative optogenetic tools that can compensate for the shortcomings of *Cr*ChRs or display stronger photo-inducing activity than *Cr*ChRs (*Hososhima et al., 2020*; *Tashiro et al., 2021*). In addition to these, the activity of several ChRs, including *Co*ChR and ChrimsonR, has been studied in zebrafish neurons, but these have not been compared directly with *Gt*CCR4 or *Kn*ChR (*Antinucci et al., 2020*).

Among the microbial enzymorhodopsins (*Mukherjee et al., 2019*; *Tsunoda et al., 2021*), *Be*GC1 is a rhodopsin guanylyl cyclase (Rh-GC) derived from the aquatic fungus *Blastocladiella emersonii* and is responsible for its zoospore phototaxis (*Avelar et al., 2014*). *Be*GC1 functions as a light-activated guanylyl cyclase. *Be*GC1 shows a rapid light-triggered increase in cGMP when expressed in *Xenopus* oocytes, mammalian cell lines and neurons, and *Caenorhabditis elegans* (*Gao et al., 2015*; *Scheib et al., 2015*). Furthermore, when *Be*GC1 was co-expressed with cyclic nucleotide-gated channel (CNG) in neurons, photoactivation of *Be*GC1 depolarized the neurons and evoked behavioral responses in *C. elegans* (*Gao et al., 2015*), suggesting the feasibility of *Be*GC1-mediated optogenetic control of neural functions.

In addition to enzymerhodopsins, photoactivated adenylyl cyclases (PACs) have also been used to regulate intracellular cyclic nucleotides in cells (*Iseki and Park, 2021*). PACs are flavoproteins that catalyze the production of cAMP in response to light stimulation. PACs from the sulfur bacterium *Beggiatoa* sp. (*b*PAC) (*Losi and Gärtner, 2008*) and the cynobacterium *Oscillatoria acuminata* (*Oa*PAC) (*Ohki et al., 2016*) are well characterized. Both *b*PAC and *Oa*PAC have a BLUF (sensors of blue-light using the flavin adenine nucleotide) domain and an adenylyl cyclase catalytic domain (Figure 2B). When expressed in *Escherichia coli* (*Ryu et al., 2010*), *Xenopus* oocytes, rat hippocampus neurons, and adult fruit flies (*Stierl et al., 2011*), *b*PAC acted as a light-dependent adenylyl cyclase. When *b*PAC was expressed in zebrafish interrenal cells, which is the teleost homologue of adrenal gland cells, cortisol increased in a light-dependent manner (*Gutierrez-Triana et al., 2015*). *b*PAC was also used for light-dependent control of sperm motility in mice (*Jansen et al., 2015*), the release of neurotransmitter in *C. elegans* neurons (*Steuer Costa et al., 2017*), and the control of developmental

processes of *Dictyostelium discoideum* (*Tanwar et al., 2017*). Compared to *b*PAC, *Oa*PAC showed lower minimum photoactivity in the dark and lower maximum photoactivity upon light stimulation when expressed in HEK293 cells (*Ohki et al., 2016*). Nevertheless, *Oa*PAC induced light-dependent axon growth in rat hippocampal neurons (*Ohki et al., 2016*). These experimental findings indicate that *b*PAC and *Oa*PAC are useful optogenetic tools, although their activity in other cell types and animals is unknown. Specifically, the effectiveness of *b*PAC and *Oa*PAC in a variety of zebrafish cells remains unclear.

In this study, we expressed the ChRs *Gt*CCR4 and *Kn*ChR, enzymorhodopsin *Be*GC1, and two PACs, *b*PAC and *Oa*PAC, in hindbrain reticulospinal V2a neurons (*Kimura et al., 2013*), which are involved in the induction of swimming behavior, and in cardiomyocytes using the zebrafish Gal4-UAS system (*Asakawa et al., 2008*), and examined their optogenetic activities. Our findings suggest that the optogenetic control using these tools provides a way to analyze the function and regulation of zebrafish neurons and cardiomyocytes in vivo.

## Results

### Optogenetic activation of cultured neuronal cells by ChRs

To express rhodopsins in vivo, fluorescence markers are useful to confirm expression in target cells. We tested two methods for marking rhodopsin-expressing cells: expression as a fusion protein with a fluorescent protein, and expression of epitope (e.g. Myc and Flag)-tagged rhodopsin and fluorescent protein separately using a viral 2A (P2A) peptide system. We expressed a fusion protein of *Gt*CCR4-3.0-EYFP, which contains the membrane-trafficking signal and the endoplasmic reticulum (ER)-export signal from a Kir2.1 potassium channel (*Gradinaru et al., 2010*; *Hoque et al., 2016*), or Myc-tagged *Gt*CCR4 (*Gt*CCR4-MT) and TagCFP separately in neuronal ND7/23 cells, which are a hybrid cell line derived from rat neonatal dorsal root ganglia neurons fused with mouse neuroblastoma (*Wood et al., 1990*). We also compared the optogenetic activity of *Kn*ChR-3.0-EYFP whose biophysical properties were previously analyzed (*Tashiro et al., 2021*). Since carboxy terminal truncations of *Kn*ChR were shown to prolong the channel open lifetime and result in stronger optogenetic activity (*Tashiro et al., 2021*), a truncated *Kn*ChR containing 272 amino acids from the N-terminus was expressed as a fusion protein containing the membrane-trafficking signal, the ER-export signal, and EYFP (*Kn*ChR-3.0-EYFP). *Cr*ChR2[T159C]-mCherry, *Co*ChR-tdTomato, and ChrimsonR-tdTomato were employed as positive controls to compare the tools investigated in this study (*Antinucci et al., 2020*; *Berndt et al., 2011*). We irradiated transfected cells with 511 nm light (*Gt*CCR4-3.0-EYFP, *Gt*CCR4-MT), 469 nm light (*Kn*ChR-3.0-EYFP, *Cr*ChR2[T159C]-mCherry, *Co*ChR-tdToamto), or 590 nm light (ChrimsonR-tdTomato). As was reported with *Gt*CCR4-EGFP (*Hososhima et al., 2020*; *Shigemura et al., 2019*; *Yamauchi et al., 2017*), when *Gt*CCR4-3.0-EYFP or *Gt*CCR4-MT-P2A-TagCFP were photoactivated in ND7/23 cells, they induced a peak photocurrent that was comparable to that of *Kn*ChR-3.0-EYFP and *Cr*ChR2[T159C]-mCherry, higher than that of ChrimsonR-tdTomato, and lower than that of *Co*ChR-tdTomato (*Figure 1A and B*). Peak and steady-state photocurrents were not significantly different in cells expressing *Gt*CCR4-3.0-EYFP and *Gt*CCR4-MT (*Figure 1B*), suggesting that photoactivation of *Gt*CCR4 from both constructs became immediately saturated. Channel-closing kinetics after shutting-off light ($\tau_{off}$) was equally fast for *Gt*CCR4-3.0-EYFP, *Gt*CCR4-MT, *Cr*ChR2[T159C]-mCherry, and *Co*ChR-tdTomato, and was faster than that observed for *Kn*ChR-3.0-EYFP, but slower than that of ChrimsonR-tdTomato (*Figure 1C*). Spectrum analysis revealed that *Gt*CCR4-3.0-EYFP responded to light of slightly longer wavelengths than *Kn*ChR-3.0-EYFP and *Cr*ChR2[T159C]-mCherry, but to shorter wavelengths for ChrimsonR-tdTomato (*Figure 1D*). The half-saturation maximum ($EC_{50}$) of peak and steady-state photocurrents was lower in *Gt*CCR4-3.0-EYFP than in *Cr*ChR2[T159C]-mCherry and ChrimsonR-tdTomato, but comparable to that of *Co*ChR-tdTomato (*Figure 1E*, *Figure 1—figure supplement 1*), suggesting that *Gt*CCR4-3.0-EYFP responded to weaker light stimulation than *Cr*ChR2[T159C]-mCherry and ChrimsonR-tdTomato. These data indicate that *Gt*CCR4-3.0-EYFP, *Gt*CCR4-MT, and *Kn*ChR-3.0-EYFP are highly sensitive tools comparable to *Co*ChR-3.0-EYFP, *Cr*ChR2[T159C]-mCherry and ChrimsonR-tdTomato. In addition, *Gt*CCR4-3.0-EYFP and *Gt*CCR4-MT in cultured cells can be activated by light of wavelengths between those of *Cr*ChR2[T159C]-mCherry and ChrimsonR-tdTomato.

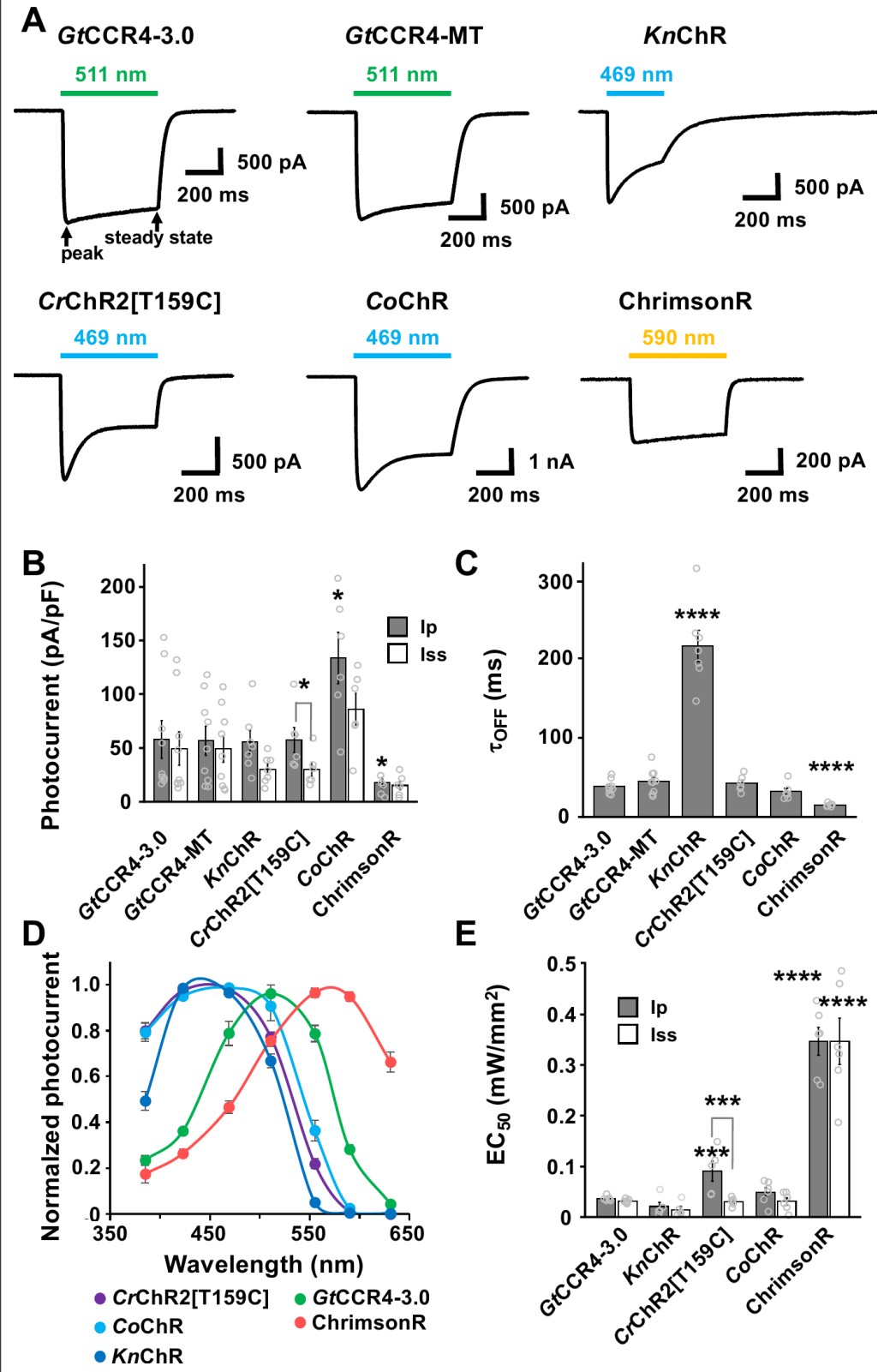

**Figure 1.** Photocurrent properties of channelrhodopsins. (**A**) Representative photocurrent traces of *Gt*CCR4-3.0-EYFP (*Gt*CCR4-3.0), *Gt*CCR4-MT-P2A-TagCFP (*Gt*CCR4-MT), *Kn*ChR, *Cr*ChR2[T159C], *Co*ChR, and ChrimsonR. Electrophysiological recordings were performed. Membrane voltage was clamped at –60 mV. Illumination sources were 511 nm light (*Gt*CCR4-3.0, *Gt*CCR4-MT), 469 nm light (*Kn*ChR, *Cr*ChR2[T159C], *Co*ChR), and 590 nm light

*Figure 1 continued*

(ChrimsonR) at 1.4 mW/mm$^2$. (**B**) Photocurrent amplitude. Gray bar: peak photocurrent (Ip); white bar: steady state photocurrent (Iss) (n = 6–9). Wilcoxon rank-sum test (*Cr*ChR2[T159C]-mCherry Ip vs. Iss, p=0.025; *Co*ChR-tdTomato Ip vs. *Gt*CCR4-3.0-EYFP Ip, p=0.025; ChrimsonR Ip vs. *Gt*CCR4-3.0-EYFP Ip, p=0.049). (**C**) Comparison of the channel closing kinetics after shutting-off light ( $\tau$ off) (n = 6–9), Wilcoxon rank-sum test (*Kn*ChR-EYFP vs. *Gt*CCR4-3.0-EYFP, p=0.0002; ChrimsonR vs. *Gt*CCR4-3.0-EYFP, p=0.0004). (**D**) The action spectrum of *Gt*CCR4-3.0 (green circle), *Kn*ChR (blue circle), *Cr*ChR2[T159C] (purple circle), *Co*ChR (light blue circle), and ChrimsonR (red circle). Illumination sources were 385, 423, 469, 511, 555, 590, or 631 nm light at 1.4 mW/mm$^2$ (*Gt*CCR4-3.0, *Cr*ChR2[T159C]) or 0.14 mW/mm$^2$ (*Kn*ChR, *Co*ChR, ChrimsonR) (n = 5–10). (**E**) Half saturation maximum (EC$_{50}$) of the peak photocurrent (gray bar) and the steady-state photocurrent (white bar) are shown (n = 5, 6), Wilcoxon rank-sum test (*Cr*ChR2[T159C]-mcherry Ip vs. Iss, p=0.0079; *Cr*ChR2[T159C]-mCherry Ip vs. *Gt*CCR4-3.0-EYFP Ip, p=0.0086; ChrimsonR Ip vs. *Gt*CCR4-3.0-EYFP Ip, p=0.0021; ChrimsonR Iss vs. *Gt*CCR4-3.0-EYFP Iss, p=0.0021). *p<0.05, **p<0.01, ***p<0.001, ****p<0.0005, Mean and SEM are indicated.

The online version of this article includes the following source data and figure supplement(s) for figure 1:

**Source data 1.** Data for *Figure 1*, photocurrent properties of ChRs.

**Figure supplement 1.** Light power dependency of photocurrent amplitude of *Gt*CCR4-3.0 (**A**), *Kn*ChR (**B**), *Cr*ChR2[T159C] (**C**), *Co*ChR (**D**), and ChrimsonR (**E**).

**Figure supplement 1—source data 1.** Data for *Figure 1—figure supplement 1*, light power dependencies of photo current amplitude of ChRs.

## Optogenetic activation of zebrafish locomotion circuit by *Gt*CCR4 and *Kn*ChR

We expressed *Gt*CCR4 and *Kn*ChR in the hindbrain reticulospinal V2a neurons of zebrafish, which were reported to drive locomotion (*Kimura et al., 2013*) by using a Gal4-UAS system (*Figure 2A*). We compared the activities of these ChRs with those of *Cr*ChR2[T159C]-mCherry, *Co*ChR-tdTomato, and ChrimsonR-tdTomato. We crossed a transgenic (Tg) zebrafish *Tg(vsx2:GAL4FF)*, which is also known as *Tg(chx10:GAL4)* and expresses a modified transcription factor Gal4-VP16 in the hindbrain reticulo-spinal V2a neurons (*Kimura et al., 2013*), with Tg lines carrying optogenetic tools that are expressed under the control of 5xUAS (upstream activating sequences of yeast *Gal1* gene) and the zebrafish *hsp70l* promoter (*Muto et al., 2017*). Since transgene-mediated protein expression depends on the nature of the introduced gene, as well as the transgene-integrated sites and copy number, we established multiple Tg lines and analyzed stable Tg lines (F$_1$ or later generations) with the highest tool expression for each tool. The expression of the fusion proteins, composed of a ChR and a fluorescent protein, in the hindbrain reticulospinal V2a neurons were detected under an epifluorescent stereomicroscope. Despite differences in expression between lines, immunohistochemistry with anti-fluorescent protein (anti-GFP and DsRed antibodies for EYFP and RFP/mCherry, respectively) and anti-MT antibodies revealed that in the reticulospinal V2a neurons, *Gt*CCR4-3.0-EYFP, *Kn*ChR-3.0-EYFP, and *Cr*ChR2[T159C]-mCherry were similarly expressed, and *Gt*CCR4-MT was expressed more strongly (*Figure 2C*, *Table 1*). *Kn*ChR-3.0-EYFP demonstrated mosaic expression (*Figure 2C*). The detection of fluorescence in *Co*ChR-tdTomato and ChrimsonR-tdTomato indicated that these ChRs were also expressed adequately in the reticulospinal V2a neurons (*Figure 2C*). We irradiated a hindbrain area of 3 days post fertilization (dpf) Tg larvae expressing *Gt*CCR4-3.0-EYFP, *Gt*CCR4-MT, *Kn*ChR-3.0-EYFP, *Cr*ChR2[T159C]-mCherry, *Co*ChR-tdTomato, and ChrimsonR-tdTomato with light of 520 nm (*Gt*CCR4, ChrimsonR) and 470 nm (*Kn*ChR, *Cr*ChR2[T159C], and *Co*ChR) for 100 ms (*Figures 2A and D and 3A–E*, *Table 1*, *Figure 2—videos 1–6*). We measured the rate at which light stimulation induced tail movements (locomotion rate, *Figure 3A*, *Figure 3—figure supplement 3*), the time from stimulation to the onset of tail movements (latency, *Figure 3B*, *Figure 3—figure supplement 2*), the duration of tail movements (*Figure 3C*), and the amplitude of tail movements (*Figure 3D*). To examine photo-sensitivity of these ChRs, we also irradiated them with light of various intensities and examined corresponding tail movements (*Figure 3E*, *Figure 3—figure supplements 1–3*).

Light stimulation of the reticulospinal V2a neurons with *Cr*ChR2[T159C]-mCherry immediately evoked tail movements (locomotion rate 67.4 ± 11.8%, latency 0.109 ± 0.0311 s, *Figure 3A and B*, *Figure 2—video 1*). Light stimulation with *Gt*CCR4-3.0-EYFP and *Gt*CCR4-MT evoked tail movements at comparable locomotion rates, although it took more time than *Cr*ChR2[T159C]-mCherry (*Gt*CCR4-3.0-EYFP locomotion rate 62.5 ± 8.77%, latency 1.59 ± 0.536 s; *Gt*CCR4-MT locomotion rate 50 ±

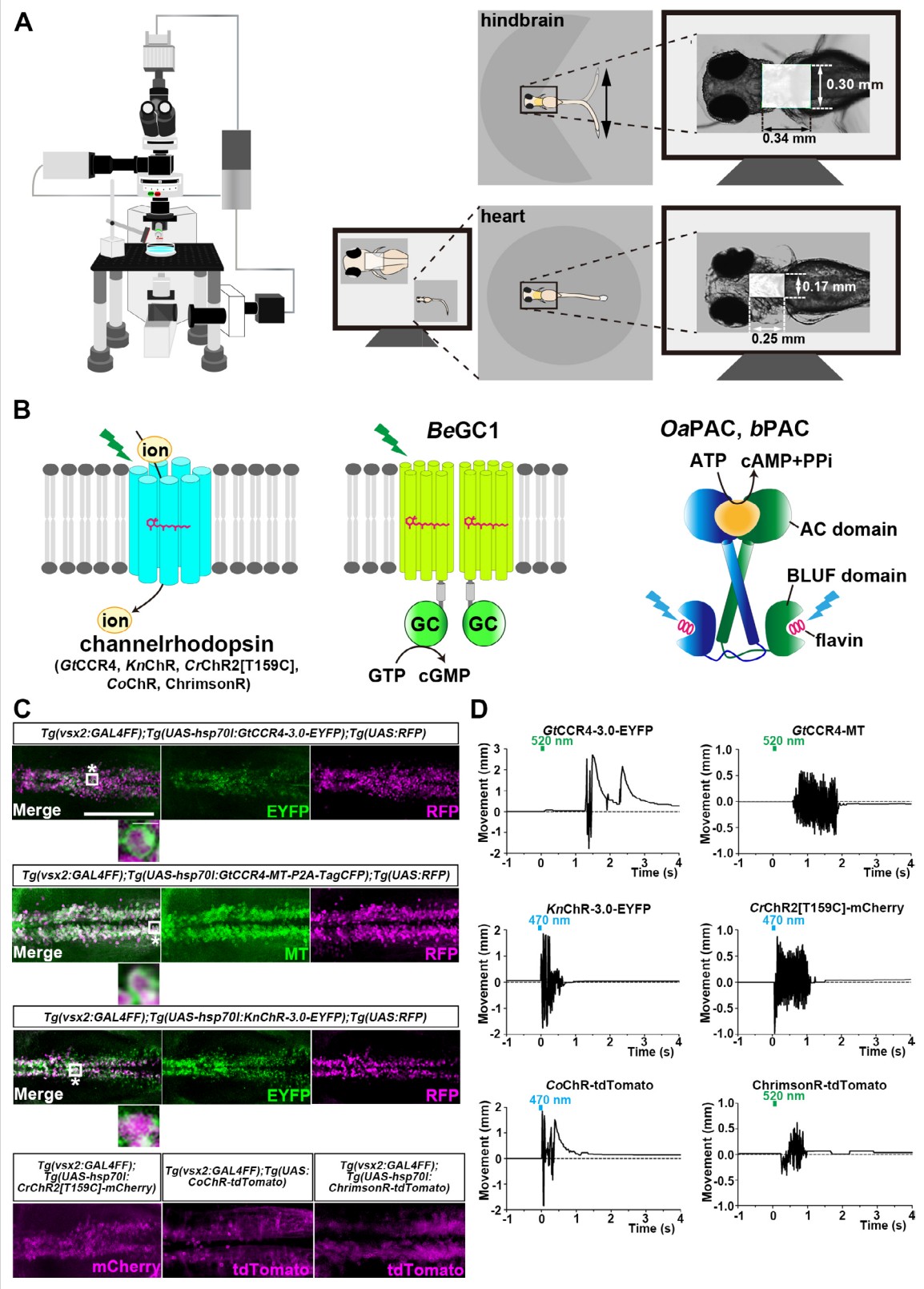

**Figure 2.** Optogenetic activation of hindbrain reticulospinal V2a neurons and cardiomyocytes by channelrhodopsins. (**A**) Schematic of experimental devices. A larva was embedded in agarose. The hindbrain region or the heart were irradiated with light. Tail (caudal fin) movements and heartbeats were monitored by a high-speed camera with infrared light. (**B**) Schematic diagram of optogenetic tools used in this study. GC, guanylyl cyclase; AC, adenylyl cyclase; BLUF, sensors of blue-light using FAD. (**C**) Expression of *Gt*CCR4-3.0-EYFP, *Gt*CCR4-MT, *Kn*ChR-3.0-EYFP, *Cr*ChR2[T159C]-mCherry,

*Figure 2 continued on next page*

*Figure 2 continued*

*Co*ChR-tdTomato, and ChrimsonR-tdTomato in the zebrafish hindbrain reticulospinal V2a neurons. 3 dpf (day post fertilization) *Tg(vsx2:GAL4FF);Tg(UAS-hsp70l:GtCCR4-3.0-EYFP, GtCCR4-MT-P2A-TagCFP, KnChR-3.0-EYFP, or CrChR2[T159C]-mCherry, myl7:mCherry)* larvae were fixed and stained with anti-GFP (EYFP, green), anti-Myc tag (green) or anti-DsRed (RFP, magenta) antibodies. For *Co*ChR and ChrimsonR, fluorescent images of the hindbrain of *Tg(vsx2:GAL4FF);Tg(UAS:CoChR-tdTomato, or UAS-hsp70l:ChrimsonR-tdTomato, myl7:mCherry)* larvae are shown. Inset: higher magnification images for the boxed areas showing double-labeled neurons. (**D**) Tail movements of 3-dpf Tg larvae expressing *Gt*CCR4-3.0-EYFP, *Gt*CCR4-MT, *Kn*ChR-3.0-EYFP, and *Cr*ChR2[T159C]-mCherry, *Co*ChR-tdTomato, and ChrimsonR-tdTomato in the reticulospinal V2a neurons after stimulation of the hindbrain area with LED (0.4 mW/mm$^2$) light with a wavelength of 520 nm (*Gt*CCR4-3.0-EYFP, *Gt*CCR4-MT, ChrimsonR-tdTomato) and 470 nm (*Kn*ChR-3.0-EYFP, *Cr*ChR2[T159C]-mCherry, *Co*ChR-tdTomato) for 100 ms. Light stimulations started at time 0 s. Typical examples are shown. Scale bars = 150 μm in (**C**), 5 μm in the insets of (**C**).

The online version of this article includes the following video and source data for figure 2:

**Source data 1.** Data for *Figure 2D*, tail movements of Tg larvae expressing ChRs.

**Figure 2—video 1.** Tail movements in a larva expressing *Cr*ChR2[T159C]-mCherry in reticulospinal V2a neurons.
https://elifesciences.org/articles/83975/figures#fig2video1

**Figure 2—video 2.** Tail movements in a larva expressing *Gt*CCR4-3.0-EYFP in reticulospinal V2a neurons.
https://elifesciences.org/articles/83975/figures#fig2video2

**Figure 2—video 3.** Tail movements in a larva expressing *Gt*CCR4-MT-P2A-TagCFP in reticulospinal V2a neurons.
https://elifesciences.org/articles/83975/figures#fig2video3

**Figure 2—video 4.** Tail movements in a larva expressing *Kn*ChR-3.0-EYFP in reticulospinal V2a neurons.
https://elifesciences.org/articles/83975/figures#fig2video4

**Figure 2—video 5.** Tail movements in a larva expressing *Co*ChR-tdTomato in reticulospinal V2a neurons.
https://elifesciences.org/articles/83975/figures#fig2video5

**Figure 2—video 6.** Tail movements in a larva expressing ChrimsonR-tdTomato in reticulospinal V2a neurons.
https://elifesciences.org/articles/83975/figures#fig2video6

8.91%, latency 1.16 ± 0.445 s, *Figure 3A and B*, *Figure 3—figure supplement 1*, *Figure 2—videos 2 and 3*). Light stimulation with these ChRs induced transient tail movements (duration: *Gt*CCR4-3.0-EYFP 0.698 ± 0.263 s, *Gt*CCR4-MT 1.51 ± 0.414 s, *Cr*ChR2[T159C]-mCherry 2.17 ± 1.19 s, *Figure 3C*). The strength of the tail movements was not significantly different between *Gt*CCR4-3.0-EYFP/*Gt*CCR4-MT and *Cr*ChR2[T159C]-mCherry (*Gt*CCR4-3.0-EYFP 0.642 ± 0.0555, *Gt*CCR4-MT 0.605 ± 0.0796, *Cr*ChR2[T159C]-mCherry 0.404 ± 0.0629, *Figure 3D*), suggesting comparable photo-inducible activities of *Gt*CCR4 and *Cr*ChR2[T159C] in the reticulospinal V2a neurons. As the activity of *Gt*CCR4-3.0-EYFP was slightly higher than that of *Gt*CCR4-MT (*Figure 2C*), we used *Gt*CCR4-3.0-EYFP for further analysis. We found that *Kn*ChR-3.0-EYFP was a more potent tool for activating the zebrafish locomotion system than *Gt*CCR4-3.0-EYFP, *Cr*ChR2[T159C]-mCherry, and ChrimsonR-tdTomato (*Figure 3* and *Figure 3—figure supplements 1–3*). In all trials of all larvae, light stimulation of *Kn*ChR immediately evoked tail movements (locomotion rate 100 ± 0%, latency 0.0198 ± 0.00357 s, and duration 0.661 ± 0.0733 s, *Figures 2D and 3A–C*, *Figure 2—video 4*). The strength of evoked tail movements with *Kn*ChR-3.0-EYFP was stronger than that of *Gt*CCR4-3.0-EYFP, *Gt*CCR4-MT, *Cr*ChR2[T159C]-mCherry, and ChrimsonR-tdTomato, and comparable to that of *Co*ChR-tdTomato (*Figure 3D*, *Figure 3—figure supplement 3*). Stimulation with *Gt*CCR4-3.0-EYFP or *Cr*ChR2[T159C]-mCherry by light of lower intensities (0.1 mW/mm$^2$) reduced locomotion rate, while that with *Kn*ChR-3.0-EYFP or *Co*ChR-tdTomato still induced tail movements in all trials (*Figure 3E*, *Figure 3—figure supplement 3*). These data indicate that the optogenetic activity of *Kn*ChR-3.0-EYFP was as strong as that of *Co*ChR-tdTomato in zebrafish reticulospinal V2a neurons.

## Optogenetic control of zebrafish heart by *Gt*CCR4 and *Kn*ChR

We next examined the optogenetic activity of *Gt*CCR4 and *Kn*ChR in cardiomyocytes, comparing it with that of the cation ChR *Cr*ChR2[T159C], *Co*ChR, and ChrimsonR, and the anion ChR *Gt*ACR1 (*Gt*ACR1-EYFP), which enables the induction of hyperpolarization in cells (*Govorunova et al., 2015*). We expressed these ChRs in zebrafish cardiomyocytes by using *Tg(myl7:GAL4FF)*, in which GAL4FF was expressed under the promoter of the cardiac myosin light chain gene *myl7*, and the UAS Tg lines. We established multiple Tg lines for each tool and used Tg lines with the highest tool expression level in cardiomyocytes. The expression of the fusion proteins, composed of ChR and fluorescent protein, in

**Table 1.** Optogenetic tools.

Microbial optogenetic tools were expressed in the hindbrain reticulospinal V2a neurons or cardiomyocytes using the Gal4-UAS system.

The expression levels of the tools were determined by immunostaining with anti-tag (MT or Flag) antibodies or anti-fluorescent marker antibodies (anti-GFP and anti-DsRed for EYFP/EGFP and mCherry, respectively) (+, weak; ++, medium; +++, strong expression). The light stimulus-dependent responses (induced swimming or cardiac arrest) are indicated by the percentage of fish that responded to light stimuli. As controls, the responses of sibling larvae that did not express the tools were also examined. ND, not determined.

| | | | | | V2a neurons | | Heart | |
| Type | Tool name | Origin | Detection | Stimulation light (nm) | Expression | Swimming response (control) | Expression | Cardiac response (control) |
|---|---|---|---|---|---|---|---|---|
| *Channelrhodopsin* | | | | | | | | |
| Cation | *Gt*CCR4-3.0-EYFP | *Guillardia theta* | EYFP fusion | 520 | ++ | 62.5%, n = 8 (14.6%, n = 8) | +++ | 100%, n = 4 (0%, n = 4) |
| Cation | *Gt*CCR4-MT | *Guillardia theta* | MT-P2A-TagCFP | 520 | +++ | 50.0%, n = 8 (8.33%, n=8) | ND | ND |
| Cation | *Kn*ChR-3.0-EYFP | *Klebsormidium nitens* | EYFP fusion | 470 | ++* | 100%, n = 8 (8.33%, n = 8) | +++ | 100%, n = 4 (0%, n = 4) |
| Cation | *Cr*ChR2 [T159C]-mCherry | *Chlamydomonas reinhardtii* | mCherry fusion | 470 | ++ | 67.4%, n = 12 (25.0%, n = 8) | + ¶ | 100%, n = 4 (0%, n = 4) |
| Cation | *Co*ChR-tdTomato | *Chloromonas oogama* | tdTomato fusion | 470 | +++† | 100%, n = 8 (2.08%, n = 8) | ND | 100%, n = 4 |
| Cation | ChrimsonR-tdTomato | *Chlamydomonas noctigama* | tdTomato fusion | 520 | ++† | 23.8%, n = 8 (6.25%, n = 8) | ND | 100%, n = 4 |
| Anion | *Gt*ACR1-EYFP | *Guillardia theta* | EYFP fusion | 520 | ++ ‡ | 80.5%, n = 6 § (13.2%, n = 6) | +++ | 100%, n = 4 (0%, n = 4) |
| *Enzymorhodopsin* | | | | | | | | |
| Guanyly cyclase | *Be*GC1-EGFP | *Blastocladiella emersonii* | EGFP fusion | 520 | +++ | 53.5%, n = 8 (10.4%, n = 8) | +++** | 0%, n = 102†† (ND) |
| *Photoactivated adenylyl cyclase* | | | | | | | | |
| Adenylyl cyclase | *b*PAC-MT | *Beggiatoa* | MT-T2A-tDimer | 470 | ++ | 60.6%, n = 8 (26.2%, n = 8) | +++ | 100%, n = 4 ‡‡ (ND) |
| Adenylyl cyclase | *Oa*PAC-Flag | *Oscillatoria acuminata* | Flag-P2A-TagCFP | 470 | ++ | 42.5%, n = 8 (15.4%, n = 8) | +++ | 0%, n = 104 §§ (ND) |

*Expression of *Kn*ChR-3.0-EYFP was mosaic.

†Expression of *Co*ChR-tdTomato and ChrimsonR-tdTomato was detected by observation with an epifluorescent stereomicroscope.

‡Expression was confirmed by detecting EYFP.

§The percentages of spontaneous tail movements elicited by white light that was inhibited by rhodopsin activation (locomotion-inhibition trials) are indicated (no rhodopsin activation was used as the control).

¶Expression of *Cr*ChR2 [T159C]-mCherry was detected by qPCR.

**The expression of *Be*GC1-EGFP was determined by observation with an epifluorescent microscope MZ16 FA and a fluorescence detection filter (460–500 nm, Leica).

††Cardiac arrest was not induced with 490–510 nm, 530–560 nm (epifluorescent stereomicroscope-equipped light source, n = 100), or 520 nm (LED) light stimuli (n = 2).

‡‡Light stimulation with 470 nm LED light for 5 s induced bradycardia, which took a few minutes to return to normal heartbeats.

§§Stimulation with 460–500 nm (epifluorescent stereomicroscope-equipped light source, n = 100) or 470 nm LED light (n = 4) induced neither cardiac arrest nor bradycardia, while stimulation with 470 nm LED light occasionally induced transient tachycardia for a few seconds (n = 2).

cardiomyocytes was detected under an epifluorescent stereomicroscope. Immunostaining with anti-fluorescent protein antibody revealed comparable expression of *Gt*CCR4-3.0-EYFP, *Kn*ChR-3.0-EYFP, and *Gt*ACR1-EYFP in 4-dpf Tg larvae (*Figure 4A*, *Table 1*). Stimulation of the entire heart area of 4-dpf Tg larvae expressing ChRs with light (520 nm for *Gt*CCR4, *Gt*ACR1, and ChrimsonR; 470 nm for *Kn*ChR, *Cr*ChR2[T159C], and *Co*ChR) for 100 ms induced cardiac arrest in all six trials, with some differences in latency (*Figures 4B and C and 5A and B*, *Figure 4—videos 1–6*). The latency of cardiac

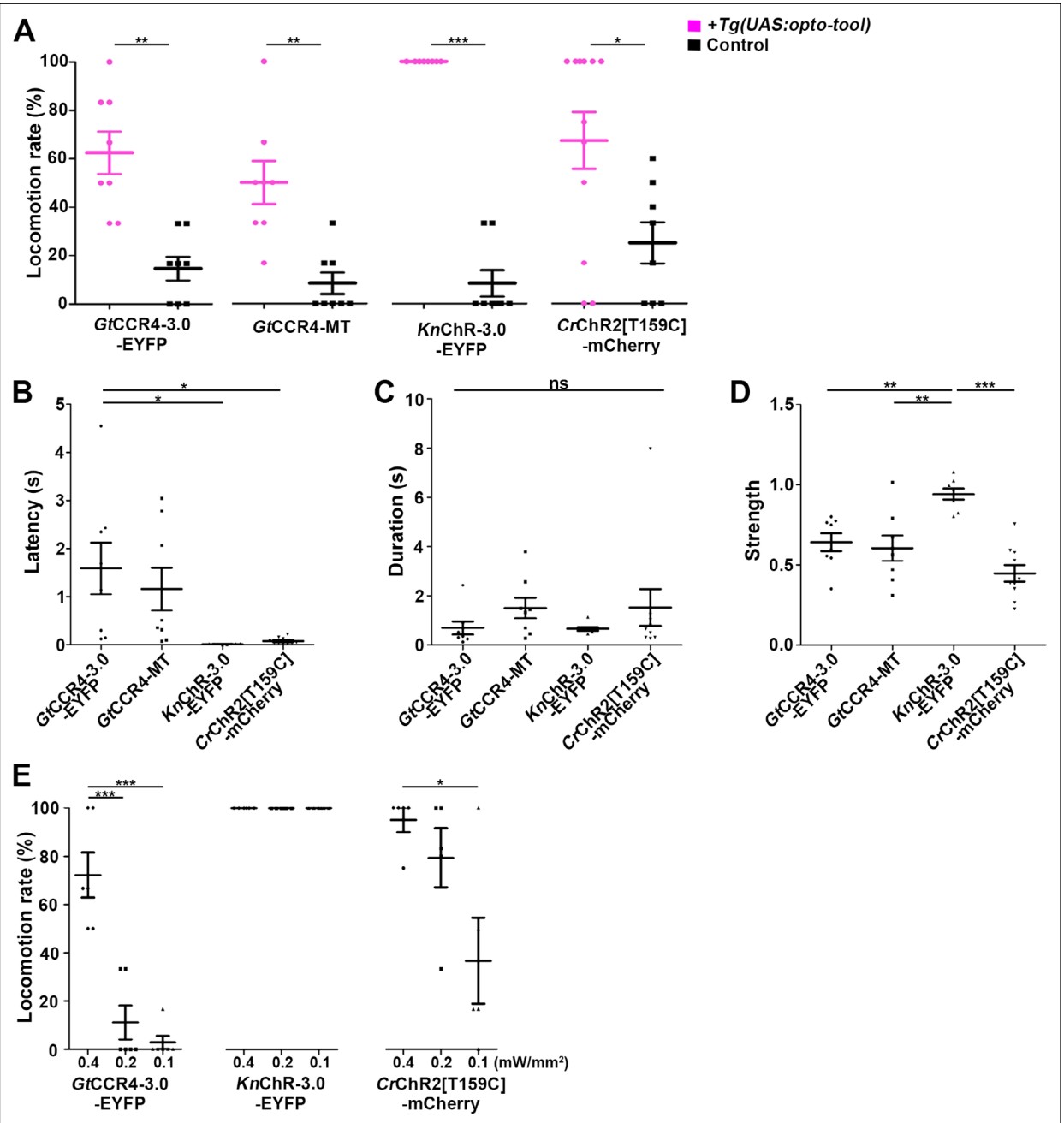

**Figure 3.** Optogenetic activation of hindbrain reticulospinal V2a neurons by *Gt*CCR4, *Kn*ChR, and *Cr*ChR2[T159C]. (**A**) Light stimulation-dependent locomotion rates of 3-dpf Tg larvae expressing *Gt*CCR4-3.0-EYFP, *Gt*CCR4-MT, *Kn*ChR-3.0-EYFP, *Cr*ChR2[T159C]-mCherry. The hindbrain area was irradiated with light (0.4 mW/mm$^2$) with a wavelength of 520 nm (*Gt*CCR4-3.0-EYFP and *Gt*CCR4-MT) or 470 nm (*Kn*ChR-3.0-EYFP and *Cr*ChR2[T159C]-mCherry) for 100 ms. Six consecutive stimulation trials were analyzed for eight rhodopsin-expressing and non-expressing (control) larvae of each Tg line. The average locomotion rates for each larva are shown, Wilcoxon rank-sum test (*Gt*CCR4-3.0-EYFP vs. control, p=0.00166; *Gt*CCR4-MT vs. control, p=0.00216; *Kn*ChR-EYFP vs. control, p=0.000266; *Cr*ChR2[T159C]-mCherry vs. control, p=0.0246). (**B–D**) Latency (**B**), duration (**C**), and strength (**D**) of tail movements. The time from the start of light stimulation to the first tail movement was defined as latency (s), and the time from the start of the first tail movement to the end of that movement was defined as duration (s). The maximum distance that the caudal fin moved from the midline divided by body length was measured as representative of its strength. One-way ANOVA with Tukey's post hoc test (latency *Gt*CCR4-3.0-EYFP vs. C*r*ChR2[T159C]-mCherry, p=0.0115; *Gt*CCR4-3.0-EYFP vs. *Kn*ChR-EYFP, p=0.0128; strength *Gt*CCR4-3.0-EYFP vs. *Kn*ChR-EYFP, p=0.00601; *Gt*CCR4-MT vs. *Kn*ChR-EYFP, p=0.00181; *Kn*ChR-EYFP vs. *Cr*ChR2[T159C]-mCherry, p=4.00e-06). (**E**) Locomotion evoked by light of various light intensities. The hindbrain area was irradiated with light at 0.4, 0.2, or 0.1 mW/mm$^2$. Six consecutive trials were analyzed for 4–6 rhodopsin-expressing larvae for each Tg (n = 6 for *Gt*CCR4-3.0-EYFP and *Kn*ChR-3.0-EFYP; n = 5 for *Cr*ChR2[T159C]-mCherry). Light stimulation experiments at 0.2 and 0.1 mW/mm$^2$ were conducted only on larvae that exhibited evoked locomotion three or more times in response to the initial light stimulation at 0.4 mW/mm$^2$. One-way ANOVA with Tukey's post hoc

*Figure 3 continued on next page*

*Figure 3 continued*

test ($Gt$CCR4-3.0-EYFP: 0.4 mW/mm$^2$ vs. 0.1 mW/mm$^2$, p=1.03e-05, 0.4 mW/mm$^2$ vs. 0.2 mW/mm$^2$, p=4.39e-05; $Cr$ChR2[T159C]-mCherry: 0.4 mW/mm$^2$ vs. 0.1 mW/mm$^2$, p=0.0185). *p<0.05, **p<0.01, ***p<0.001, ns: not significant. Means and SEMs are indicated.

The online version of this article includes the following source data and figure supplement(s) for figure 3:

**Source data 1.** Data for *Figure 3*, optogenetic activation of hindbrain reticulospinal V2a neurons by ChRs.

**Figure supplement 1.** Latency of locomotion in ChR-expressing and non-expressing larvae.

**Figure supplement 1—source data 1.** Data for *Figure 3—figure supplement 1*, latency of locomotion in ChR-expressing and non-expressing larvae.

**Figure supplement 2.** Latency and duration of locomotion induced with channelrhodopsins (ChRs) by light of various light intensities.

**Figure supplement 2—source data 1.** Data for *Figure 3—figure supplement 2*, latency and duration of locomotion induced with ChRs by light of various intensities.

**Figure supplement 3.** Optogenetic activation of hindbrain reticulospinal V2a neurons by *Co*ChR and ChrimsonR.

**Figure supplement 3—source data 1.** Data for *Figure 3—figure supplement 3*, optogenetic activation of V2a neurons by *Co*ChR and ChrimsonR.

arrest induced by stimulation with these ChRs was short, especially that of *Kn*ChR-3.0-EYFP, which was shorter than that of *Gt*ACR1-EYFP (*Gt*CCR4-3.0-EYFP 163 ± 12 ms, *Kn*ChR-3.0-EYFP 55.2 ± 13.4 ms, *Cr*ChR2[T159C]-mCherry 112 ± 8.72 ms, *Gt*ACR1-EYFP 243 ± 48.8 ms, *Co*ChR-tdTomato 215 ± 31.4 ms, ChrimsonR-tdTomato 212 ± 55.2 ms, *Figure 5B*, *Figure 5—figure supplement 2*).

Heartbeats (HBs) resumed within 2 s after light stimulation but took longer when stimulated with *Gt*ACR1-EYFP than with *Gt*CCR4-3.0-EFYP or *Kn*ChR-3.0-EYFP (*Gt*CCR4-3.0-EYFP 418 ± 91.4 ms, *Kn*ChR-3.0-EYFP 742 ± 62.3 ms, *Cr*ChR2[T159C]-mCherry 284 ± 29.7 ms, *Gt*ACR1-EYFP 1350 ± 215 ms, *Co*ChR-tdTomato 379 ± 45.6 ms, ChrimsonR-tdTomato 319 ± 41.7 ms, *Figure 5C*, *Figure 5—figure supplement 2*). Stimulation with *Gt*CCR4-3.0-EYFP by light of lower light intensity (0.05 mW/mm$^2$) reduced cardiac arrest rate, while stimulation with *Kn*ChR-3.0-EYFP or *Co*ChR-tdTomato still induced cardiac arrest in all trials (*Figure 5D*, *Figure 5—figure supplements 1 and 2*). These data again indicate that optogenetic activity of *Kn*ChR-3.0-EYFP is as strong as that of *Co*ChR-tdTomato in zebrafish cardiomyocytes. Considering that *Gt*ACR1 is an anion ChR and *Gt*CCR4/*Kn*ChR are cation ChRs, the mechanism of cardiac arrest is expected to be different. To address this issue, we monitored intracellular Ca$^{2+}$ concentration in cardiomyocytes using GCaMP6s. Light stimulation with *Kn*ChR-3.0-EYFP increased fluorescence intensity ($\Delta$F/F) of GCaMP6s in the heart and induced cardiac muscle contraction (*Figure 5E*, *Figure 5—video 1*), whereas that with *Gt*ACR1-EYFP reduced the fluorescence intensity of GCaMP6s and induced relaxation of the myocardium (*Figure 5F*, *Figure 5—video 2*), suggesting distinct mechanisms for cardiac arrest induced by the cation ChRs (*Gt*CCR4 and *Kn*ChR) and anion ChR (*Gt*ACR1).

## Optogenetic control of cAMP/cGMP by *Be*GC1 and PACs

We examined the optogenetic activity of the fungal guanylyl cyclase rhodopsin *Be*GC1 and compared it to that of the flavoprotein-type bacterial PACs *b*PAC and *Oa*PAC. By using *Tg(vsx2:GAL4FF)* and UAS Tg lines, we established Tg lines that expressed *Be*GC1-EGFP, *b*PAC-MT, and *Oa*PAC-Flag in the reticulospinal V2a neurons (*Figures 6A and 7A*). Although the levels of expression of *b*PAC-MT and *Oa*PAC-Flag were slightly lower than that of *Be*GC1, light stimulation of the reticulospinal V2a neurons with *Be*GC1-EGFP (520 nm), *b*PAC-MT (470 nm), and *Oa*PAC-Flag (470 nm) at 0.4 mW/mm$^2$ for 500 ms evoked relatively high-frequency tail movements (*Be*GC1-EGFP 53.5 ± 7.25%, *b*PAC-MT 60.6 ± 9.8%, *Oa*PAC-Flag 42.5 ± 13.7%, *Figures 6B and C and 7B and C*, *Figure 6—video 1*, *Figure 7—videos 1 and 2*). The tail movements induced by activation with *Be*GC1, *b*PAC, or *Oa*PAC had a long latency (*Be*GC1-EGFP 0.674 ± 0.118 s, *b*PAC-MT 2.6 ± 0.396 s, *Oa*PAC-Flag 4.0 ± 0.582 s, *Figures 6D and 7D*), but similar duration and strength compared to activation with the ChRs (duration *Be*GC1-EGFP 2.05 ± 0.33 s, *b*PAC-MT 2.02 ± 0.531 s, *Oa*PAC-Flag 2.83 ± 1 s; strength *Be*GC1-EGFP 0.815 ± 0.0926, *b*PAC-MT 0.66 ± 0.116, *Oa*PAC-Flag 0.705 ± 0.102, *Figures 6E and F and 7E and F*). The latency of light responses in Tg fish expressing *b*PAC or *Oa*PAC was long, whereas in control larvae that did not express these tools, spontaneous responses within 8 s were significantly lower (*Figure 7C*). Even when observing within 30 s after the light stimulus, the frequency of spontaneous tail movements was low (*Figure 7—figure supplement 1*), indicating that most of the tail movements induced with *b*PAC and *Oa*PAC were robust responses to light. When a cAMP fluorescent indicator Flamindo2 (*Odaka et al., 2014*) was co-expressed with *b*PAC-MT or *Oa*PAC-Flag in postmitotic neurons by the *elavl3* promoter

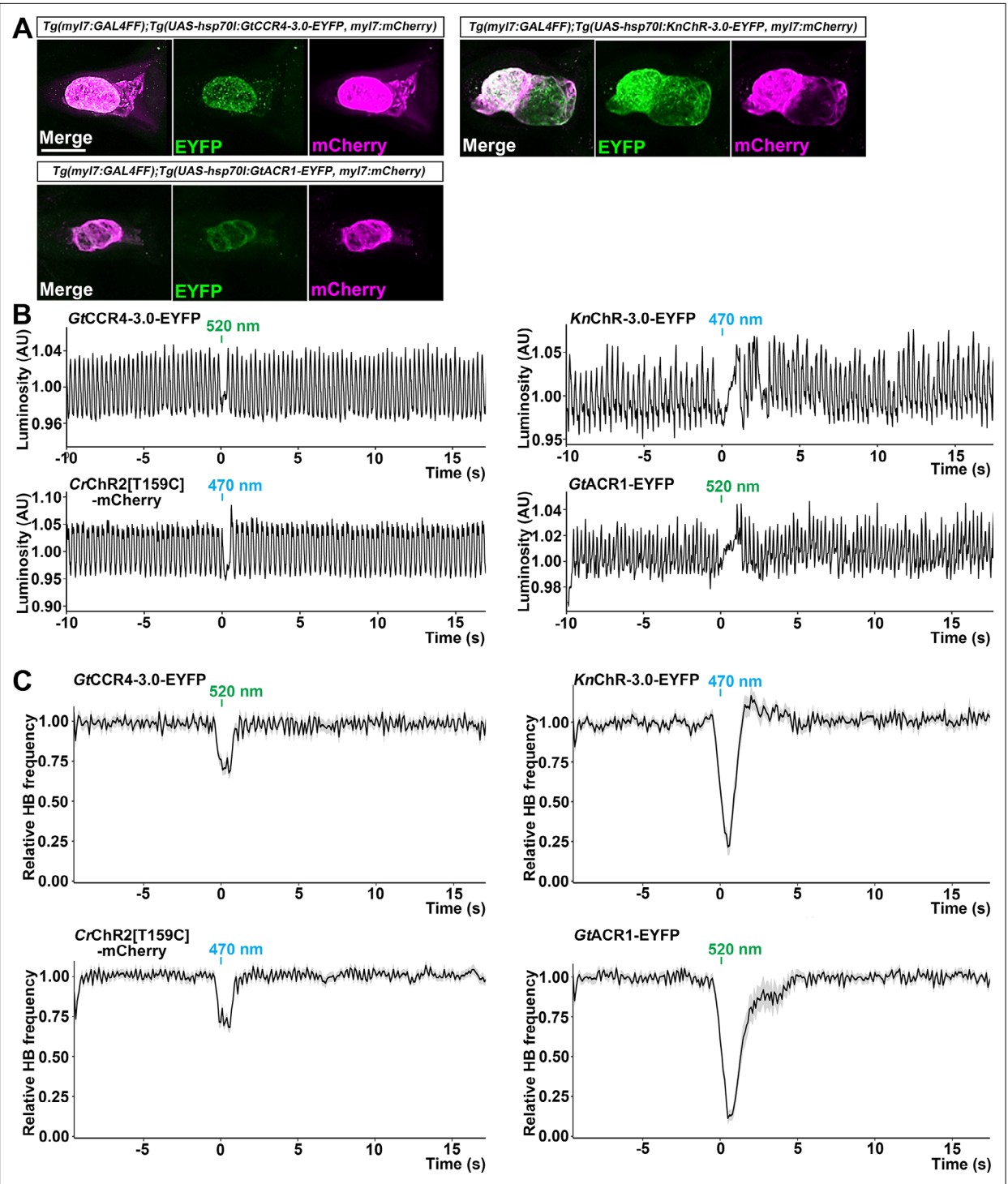

**Figure 4.** Cardiac arrest induced with *Gt*CCR4-3.0, *Kn*ChR, *Cr*ChR2[T159C], and *Gt*ACR1. (**A**) Expression of *Gt*CCR4-3.0-EYFP, *Kn*ChR-3.0-EYFP, and *Gt*ACR1-EYFP in cardiomyocytes of *Tg(myl7:GAL4FF);Tg(UAS-hsp70l:GtCCR4-3.0-EYFP, KnChR-3.0-EYFP,* or *GtACR1-EYFP, myl7:mCherry)* larvae. 4-dpf Tg larvae were fixed and stained with anti-GFP (for EYFP, green) or anti-DsRed (for mCherry, magenta) antibodies. Z stacks of confocal images. (**B**) Heartbeat (HB) monitoring by changes in luminosity (AU, arbitrary units). The entire heart area of 4-dpf Tg larvae was irradiated with light (520 nm for *Gt*CCR4 and *Gt*ACR1; 470 nm for *Kn*ChR and *Cr*ChR2) for 100 ms at a strength of 0.5 mW/mm$^2$. (**C**) Average of relative HB frequency. The heart area was irradiated at the indicated periods. Six consecutive stimulus trials were analyzed for four rhodopsin-expressing larvae of each Tg line. Relative HB frequency was calculated from HB data during 1 s and 500 ms before and after each time point, so the change in the HB frequency was observed before light stimulation, even though cardiac arrest occurred during light stimulation. Gray shading indicates SEMs. Scale bar = 100 μm in (**A**).

The online version of this article includes the following video and source data for figure 4:

*Figure 4 continued on next page*

*Figure 4 continued*

**Source data 1.** Data for *Figure 4*, cardiac arrest induced with ChRs.

**Figure 4—video 1.** Heart movements in a larva expressing *Gt*CCR4-3.0-EYFP in cardiomyocytes.
https://elifesciences.org/articles/83975/figures#fig4video1

**Figure 4—video 2.** Heart movements in a larva expressing *Kn*ChR-3.0-EYFP in cardiomyocytes.
https://elifesciences.org/articles/83975/figures#fig4video2

**Figure 4—video 3.** Heart movements in a larva expressing *Cr*ChR2[T159C]-mCherry in cardiomyocytes.
https://elifesciences.org/articles/83975/figures#fig4video3

**Figure 4—video 4.** Heart movements in a larva expressing *Gt*ACR1-EYFP in cardiomyocytes.
https://elifesciences.org/articles/83975/figures#fig4video4

**Figure 4—video 5.** Heart movements in a larva expressing *Co*ChR-tdTomato in cardiomyocytes.
https://elifesciences.org/articles/83975/figures#fig4video5

**Figure 4—video 6.** Heart movements in a larva expressing ChrimsonR-tdTomato in cardiomyocytes.
https://elifesciences.org/articles/83975/figures#fig4video6

and the *elavl3* promoter-driven GAL4-VP16 Tg line, continuous light stimulation reduced the ΔF/F of Flamindo2, which is indicative of an intracellular increase in cAMP (*Odaka et al., 2014*), in the optic tectum (*Figure 7G and H*). We also used Tg lines expressing *Be*GC1, *b*PAC or *Oa*PAC in cardiomyocytes. Although light stimulation of cardiomyocytes with *Be*GC1 or *Oa*PAC (*Table 1*) induced neither cardiac arrest nor bradycardia, activation with *b*PAC for 5 s gradually reduced HBs and it took a few minutes to return to normal HBs (*Figure 7—figure supplement 2*, *Figure 7—video 3*). These data indicate that *Be*GC1 and *Oa*PAC can be used for optogenetic activation of neurons but not cardiomyocytes, while *b*PAC can be used for optogenetic control of neurons as well as cardiomyocytes in zebrafish.

## Discussion
### Utility of ChRs *Gt*CCR4 and *Kn*ChR

As was reported for *Gt*CCR4-EGFP (*Hososhima et al., 2020*; *Shigemura et al., 2019*; *Yamauchi et al., 2017*), *Gt*CCR4-3.0-EYFP and *Gt*CCR4-MT were more sensitive to light stimuli than *Cr*ChR2[T159C]-mCherry in cultured mammalian neuronal cells (*Figure 1*). However, the optogenetic ability of *Gt*CCR4-3.0-EYFP and *Cr*ChR2[T159C]-mCherry to induce tail movements in the reticulospinal V2a neurons was comparable (*Figure 3A and D*), and *Gt*CCR4-3.0-EYFP took longer to initiate tail movements (*Figure 3B*). There are a couple of explanations for this difference. First, the expression of *Gt*CCR4-3.0-EYFP and *Cr*ChR2[T159C]-mCherry proteins might be different. Differences in levels of mRNA expression between Tg lines cannot be ruled out. In addition, there might be differences in translation, cell surface trafficking, and protein stability between these two rhodopsins in zebrafish neurons. *Gt*CCR4-3.0-EYFP contains a membrane-trafficking signal and an ER-export signal that allows expression on the cell surface, but *Gt*CCR4-3.0-EYFP proteins might aggregate slightly within the cytoplasm and might not express efficiently on the cell surface of reticulospinal V2a neurons (*Figure 2C*). Differences in ion channel properties might also affect their activity in vivo. *Cr*ChR2 is permeable to not only $Na^+$ but also $H^+$ and $Ca^{2+}$, whereas *Gt*CCR4 is relatively specific to $Na^+$ (*Nagel et al., 2003*; *Shigemura et al., 2019*). Activation of reticulospinal V2a neurons with *Cr*ChR2 induces an influx of $Na^+$, $H^+$, and $Ca^{2+}$, while activation with *Gt*CCR4 induces only an influx of $Na^+$. This may account for the difference in light-evoked tail movements between *Gt*CCR4 and *Cr*ChR2. On the other hand, the $Na^+$-specific channel property of *Gt*CCR4 may favor distinguishing depolarization effects from intracellular $Ca^{2+}$ signaling. Furthermore, the activation wavelength of *Gt*CCR4 is slightly more red-shifted than that of *Cr*ChR2 (*Figure 1D*; *Hososhima et al., 2020*; *Nagel et al., 2003*; *Shigemura et al., 2019*), which might be useful when used in conjunction with short-wavelength optogenetic tools or neural activation sensors.

We found that *Kn*ChR was a more potent optogenetic tool than *Gt*CCR4, *Cr*ChR2, and ChrimsonR in zebrafish reticulospinal V2a neurons. Optogenetic activity of *Kn*ChR was comparable to that of *Co*ChR in both reticulospinal V2a neurons and cardiomyocytes (*Figures 1, 3 and 5*). Truncation of *Kn*ChR prolonged the channel open lifetime by more than 10-fold (*Tashiro et al., 2021*; *Figure 1*).

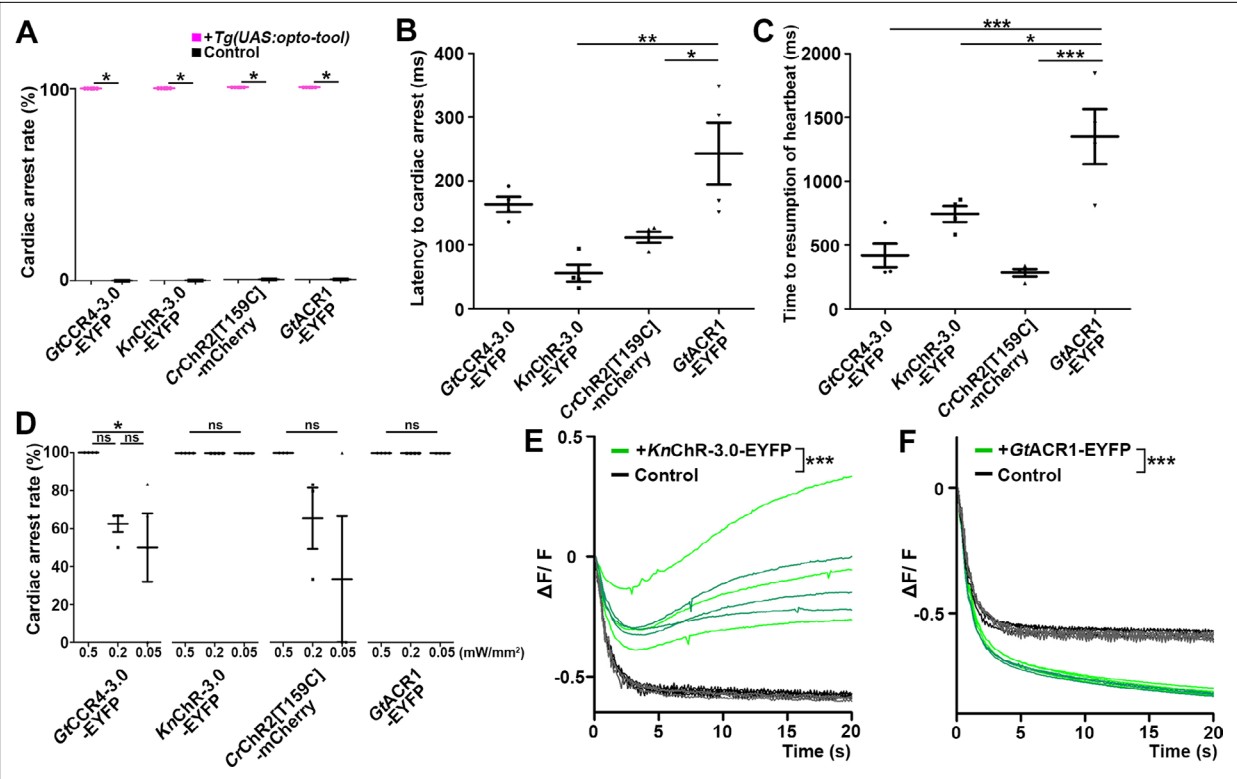

**Figure 5.** Cardiac arrest and resumption of heartbeats with GtCCR4-3.0, KnChR, CrChR2[T159C], and GtACR1. (**A**) Cardiac arrest rates of 4-dpf Tg larvae expressing GtCCR4-3.0-EYFP, KnChR-3.0-EYFP, CrChR2[T159C]-mCherry, or GtACR1-EYFP in cardiomyocytes. The heart area was irradiated with appropriate light (520 nm for GtCCR4 and GtACR1; 470 nm for KnChR and CrChR2) for 100 ms at a strength of 0.5 mW/mm². Sibling larvae that did not express the rhodopsins were used as controls. Six consecutive stimulation trials were analyzed for four rhodopsin-expressing larvae and four control larvae of each Tg line, Wilcoxon rank-sum test (GtCCR4-3.0-EYFP, KnChR-3.0-EYFP, CrChR2[T159C]-mCherry, and GtACR1-EYFP, p=0.0131). (**B, C**) Latency to cardiac arrest (**B**) and time to resumption of HBs (**C**) after light stimulation with GtCCR4-3.0-EYFP, KnChR-3.0-EYFP, CrChR2[T159C]-mCherry, or GtACR1-EYFP. HB data were obtained from the experiments described above (**A**). One-way ANOVA with Tukey's post hoc test (latency to cardiac arrest KnChR-3.0-EYFP vs. GtACR1-EYFP, p=0.00144; CrChR2[T159C]-mCherry vs. GtACR1-EYFP, p=0.0190; time to resumption of heartbeat GtCCR4-3.0-EYFP vs. GtACR1-EYFP, p=0.000786; KnChR-3.0-EYFP vs. GtACR1-EYFP, p=0.0189; CrChR2[T159C]-mCherry vs. GtACR1-EYFP, p=0.000236). (**D**) Light intensity dependence of cardiac arrest rates of 4-dpf Tg larvae expressing GtCCR4-3.0-EYFP, KnChR-3.0-EYFP, CrChR2[T159C]-mCherry, GtACR1-EYFP in cardiomyocytes. The heart area was irradiated with light (520 nm for GtCCR4 and GtACR1; 470 nm for KnChR and CrChR2) for 100 ms at a strength of 0.5, 0.2, or 0.05 mW/mm². Six consecutive stimulation trials were analyzed for four rhodopsin-expressing larvae of each Tg line. One-way ANOVA with Tukey's post hoc test (GtCCR4-3.0-EYFP: 0.5 mW/mm² vs. 0.05 mW/mm²; p=0.0222). (**E, F**) Changes in fluorescence intensity of GCaMP6s (ΔF/F) in the heart of 4-dpf Tg larvae expressing KnChR-3.0-EYFP and GCaMP6s (**E**), or GtACR1-EYFP and GCaMP6s (**F**). Sibling larvae that did not express the rhodopsins were used as the control. The heart area of Tg larvae was stimulated with a fluorescence detection filter (excitation 470–495 nm, emission 510–550 nm). Two rhodopsin-expressing larvae (green) and two control larvae (black) were analyzed for each rhodopsin. Three trials were analyzed for each larva. The linear mixed-effects model was used for statistical analysis. *p<0.05, **p<0.01, ***p<0.001, ns: not significant. Means and SEMs are indicated.

The online version of this article includes the following video, source data, and figure supplement(s) for figure 5:

**Source data 1.** Data for *Figure 5*, cardiac arrest and resumption of heartbeats with ChRs.

**Figure supplement 1.** Cardiac arrest induced with channelrhodopsins (ChRs) by light of various intensities.

**Figure supplement 1—source data 1.** Data for *Figure 5—figure supplement 1*, cardiac arrest induced with ChRs by light of various intensities.

**Figure supplement 2.** Cardiac arrest induced with CoChR and ChrimsonR.

**Figure supplement 2—source data 1.** Data for *Figure 5—figure supplement 2*, cardiac arrest induced with CoChR and ChrimsonR.

**Figure 5—video 1.** Ca²⁺ imaging in the heart of a larva expressing KnChR-3.0-EYFP and GCaMP6s in cardiomyocytes.
https://elifesciences.org/articles/83975/figures#fig5video1

**Figure 5—video 2.** Ca²⁺ imaging in the heart of a larva expressing GtACR1-EYFP and GCaMP6s in cardiomyocytes.
https://elifesciences.org/articles/83975/figures#fig5video2

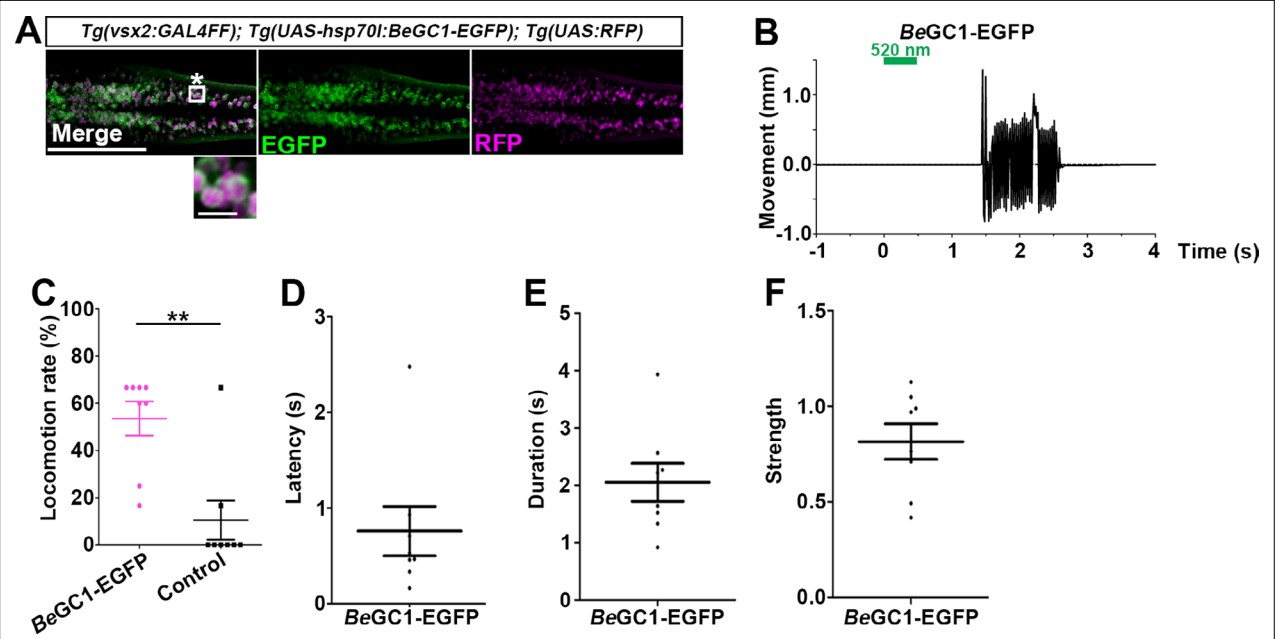

**Figure 6.** Optogenetic activation of hindbrain reticulospinal V2a neurons with *Be*GC1-EGFP. (**A**) Expression of *Be*GC1-EGFP in the hindbrain reticulospinal V2a neurons. 3-dpf *Tg(vsx2:GAL4FF);Tg(UAS:BeGC1-EGFP, myl7:mCherry);Tg(UAS:RFP)* larvae were fixed and stained with anti-GFP (EGFP, green) or anti-DsRed (RFP, magenta) antibodies. Inset: higher magnification images for the boxed area showing double-labeled neurons. In the inset, fluorescence signal intensities were modified to compare the subcellular localization of the tools. (**B**) Tail movements of 3-dpf Tg larvae expressing *Be*GC1 in the reticulospinal V2a neurons after stimulation with light (0.4 mW/mm²) at 520 nm for 500 ms. The stimulation started at time 0 s. A typical induced tail movement is shown. (**C**) Light stimulation-dependent locomotion rates of 3-dpf *Be*GC1-expressing larvae or non-expressing sibling control larvae. Six consecutive stimulation trials were analyzed for eight *Be*GC1-expressing and eight non-expressing larvae. The average locomotion rates for each larva are shown. Wilcoxon rank-sum test (*Be*GC1-EGFP vs. control, p=0.00608). (**D–F**) Latency (**D**), duration (**E**), and strength (**F**) of induced tail movements in the *Be*GC1-expressing larvae. The time from the start of light stimulation to the first tail movement was defined as latency (s), and the time from the start of the first tail movement to the end of that movement was defined as duration (s). The maximum distance that the caudal fin moved from the midline divided by body length reflected its strength. Scale bars = 150 μm in (**A**), 10 μm in the insets of (**A**). **p<0.01. Means and SEMs are indicated.

The online version of this article includes the following video and source data for figure 6:

**Source data 1.** Data for *Figure 6*, optogenetic activation of hindbrain reticulospinal V2a neurons with *Be*GC1-EGFP.

**Figure 6—video 1.** Tail movements in a larva expressing *Be*GC1-EGFP in reticulospinal V2a neurons.
https://elifesciences.org/articles/83975/figures#fig6video1

*Kn*ChR conducts various monovalent and bivalent cations, including H⁺, Na⁺, and Ca²⁺, while *Kn*ChR has a higher permeability to Na⁺ and Ca²⁺ and a higher permeability ratio of Ca²⁺ to Na⁺ than *Cr*ChR2 (*Tashiro et al., 2021*). These properties may contribute to the high photo-inducible activity of *Kn*ChR. Activation of *Kn*ChR may induce the influx of more cations with a longer channel open time than *Cr*ChR2 and ChrimsonR, leading to stronger cell depolarization. The optogenetic activity of *Kn*ChR was comparable to that of *Gt*CCR4 in cultured cells, but higher than that of *Gt*CCR4 in zebrafish reticulospinal V2a neurons and cardiomyocytes. While the exact reason is unclear, it is possible that the expression of functional *Kn*ChR protein may be high in zebrafish cells. Furthermore, since *Kn*ChR can be activated by light with a short wavelength (maximal sensitivity between 430 and 460 nm), *Kn*ChR can be used in conjunction with other red-shifted optogenetic tools and cell activity sensors.

The photoactivation of both cation (*Gt*CCR4, *Kn*ChR) and anion (*Gt*ACR1) ChRs induced cardiac arrest (*Figures 4 and 5*). However, activation of *Kn*ChR and *Gt*ACR1 increased and decreased intracellular Ca²⁺ in cardiomyocytes, respectively (*Figure 5*). Since Ca²⁺ is a readout of depolarization in cardiomyocytes, these data suggest that activation of the cation ChRs depolarized cardiomyocytes, increased intracellular Ca²⁺ concentration, and inhibited cardiac resumption. Alternatively, cardiac arrest can potentially be explained by a phenomenon known as depolarization block, in which action potentials cannot be generated because the cells remain in a depolarized state. In contrast, activation

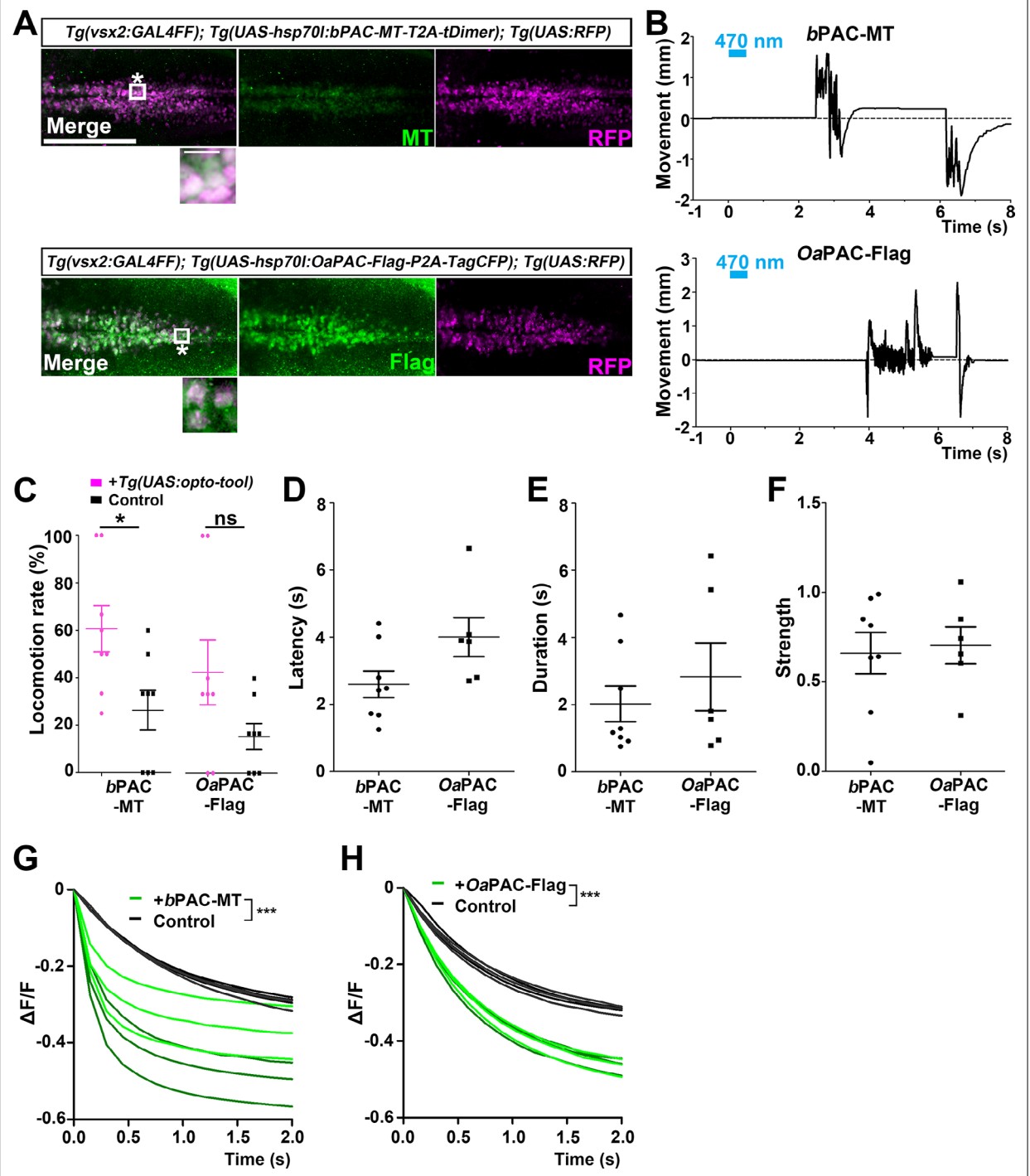

**Figure 7.** Optogenetic activation of reticulospinal V2a neurons with *b*PAC and *Oa*PAC. (**A**) Expression of *b*PAC and *Oa*PAC in reticulospinal V2a neurons. 3-dpf *Tg(vsx2:GAL4FF);Tg(UAS-hsp70l:bPAC-MT-T2A-tDimer, myl7:mCherry);Tg(UAS;RFP)* and *Tg(vsx2:GAL4FF);Tg(UAS-hsp70l:OaPAC-Flag-P2A-TagCFP, myl7:mCherry);Tg(UAS:RFP)* larvae were fixed and stained with anti-Myc or anti-Flag (green), and anti-DsRed (RFP, magenta) antibodies. To detect relatively weak fluorescent signals of *b*PAC-MT and *Oa*PAC-Flag, images were taken with increased laser power (2× for *b*PAC-MT and 4× for *Oa*PAC-Flag compared to *Be*GC1-EGFP in *Figure 6A*). Inset: higher magnification images for the boxed areas showing double-labeled neurons. (**B**) Tail movements of 3-dpf Tg larvae expressing *b*PAC-MT or *Oa*PAC-Flag in the reticulospinal V2a neurons after stimulation with light (0.4 mW/mm²) at 470 nm for 500 ms. The stimulation started at time 0 s. Typical examples are shown. (**C**) Light-induced locomotion rates. Larvae that did not express PACs were used as controls. Six consecutive stimulation trials for eight PAC-expressing and eight control larvae were analyzed. The average locomotion rates for each larva are shown. Wilcoxon rank-sum test (*b*PAC-MT vs. control, p=0.0376; *Oa*PAC-Flag vs. control, p=0.145). (**D–F**) Latency (**D**), duration (**E**), and strength (**F**) for light stimulus-induced tail movements in larvae expressing *b*PAC-MT or *Oa*PAC-Flag. The data for each larva are plotted in

*Figure 7 continued on next page*

*Figure 7 continued*

graphs. (**G, H**) Changes in fluorescence intensity (ΔF/F) of cAMP indicator Flamido2 in neurons of Tg larvae expressing *b*PAC-MT (**G**) or *Oa*PAC-Flag (**H**) after light stimulation. The entire optic tectum area of 3-dpf *Tg(elavl3:GAL4-VP16); Tg(elavl3:Flamindo2); Tg(UAS-hsp70l:bPAC-MT-T2A-tDimer)* and *Tg(elavl3:GAL4-VP16); Tg(elavl3:Flamindo2); Tg(UAS-hsp70l:OaPAC-Flag-P2A-TagCFP)* larvae was stimulated with a fluorescence detection filter (excitation 470–495 nm, emission 510–550 nm). The fluorescence intensity of the optic tectum was measured, and ΔF/F was calculated. Sibling larvae that did not express PACs were used as controls. Three trials for two PAC-expressing (green) and two control (black) larvae were analyzed, and the data from a total of six trials are plotted on graphs. The linear mixed-effects model was used. Scale bars = 150 μm in (**A**), 10 μm in insets of (**A**). *p<0.05, ***p<0.001, ns: not significant. Means and SEMs are indicated.

The online version of this article includes the following video, source data, and figure supplement(s) for figure 7:

**Source data 1.** Data for *Figure 7*, optogenetic activation of reticulospinal neurons with PACs.

**Figure supplement 1.** Latency of locomotion in PAC-expressing and non-expressing larvae.

**Figure supplement 1—source data 1.** Data for *Figure 7—figure supplement 1*, latency of locomotion in PAC-expressing and non-expressing larvae.

**Figure supplement 2.** Optogenetic control of the heart by *b*PAC or *Oa*PAC.

**Figure supplement 2—source data 1.** Data for *Figure 7—figure supplement 2*, optogenetic control of the heart by *b*PAC.

**Figure 7—video 1.** Tail movements in a larva expressing *b*PAC-MT in reticulospinal V2a neurons.
https://elifesciences.org/articles/83975/figures#fig7video1

**Figure 7—video 2.** Tail movements in a larva expressing *Oa*PAC-Flag in reticulospinal V2a neurons.
https://elifesciences.org/articles/83975/figures#fig7video2

**Figure 7—video 3.** Heart movements in a larva expressing *b*PAC-MT in cardiomyocytes.
https://elifesciences.org/articles/83975/figures#fig7video3

with the anion ChR hyperpolarized cardiomyocytes, decreased intracellular $Ca^{2+}$ concentration, and inhibited cardiac contraction. Tg larvae with a high expression of *Kn*ChR-3.0-EYFP in the heart always showed cardiac arrest after light stimulation (**Figures 4 and 5**). This finding indicates that *Kn*ChR is a strong tool. By altering the degree of depolarization by *Kn*ChR by changing the expression level or the intensity of light stimulation, the function of cardiomyocytes and other cells may be precisely controlled. Highly sensitive *Kn*ChR has the potential to identify neural circuits that have not been previously identifiable with other optogenetic tools.

In this study, we demonstrated that *Gt*CCR4, *Kn*ChR, and *Cr*ChR2 function in both reticulospinal V2a neurons and cardiomyocytes in zebrafish. Given the different ion-channel properties of *Gt*CCR4, *Kn*ChR, and *Cr*ChR2, they can be used for optogenetic manipulation of cell activities in a variety of applications in zebrafish.

## Utility of enzyme rhodopsin *Be*GC1 and bacterial flavoprotein PACs

cAMP and cGMP are major second messengers that regulate multiple biological functions in a variety of tissues. We expressed *Be*GC1 and *b*PAC/*Oa*PAC to manipulate intracellular cGMP and cAMP signaling in reticulospinal V2a neurons and cardiomyocytes (**Figures 6 and 7**, **Figure 7—figure supplements 1 and 2**). Light stimulation of the V2a neuron with *Be*GC1 as well as *b*PAC/*Oa*PAC induced tail movements with a longer delay than when stimulated with ChRs. There are two possible mechanisms by which cyclic nucleotides control cell excitability. One mechanism is through cyclic nucleotide-gated ion (CNG) channels, in which binding of cGMP or cAMP to CNG channels opens the cation channels and depolarizes the cell (**Bradley et al., 2005**; **Matulef and Zagotta, 2003**). The other is through cAMP-dependent protein kinase (PKA), in which binding of cAMP to the regulatory unit of PKA releases the catalytic unit of PKA, resulting in phosphorylation of cation channels such as the voltage-dependent $Ca^{2+}$ channel CaV1.2 (**Fu et al., 2014**; **McDonald et al., 1994**; **Reuter, 1983**). The former mechanism is often used for various sensory systems in the nervous system and the latter for the sympathetic noradrenergic regulation of HBs. Which mechanism is used may depend on the availability of necessary components to activate these mechanisms. It is likely that the CNG-mediated mechanism is involved in the activation of reticulospinal V2a neurons. In this mechanism, neurons are not activated until the intracellular cGMP/cAMP concentration reaches the threshold for CNG activation. Consistent with this, activation with *Be*GC1 and PACs induced neural activation with a relatively long delay (**Figures 6 and 7**). On the other hand, the PKA-mediated pathway may be involved in the heart. Activation of *b*PAC – but not *Be*GC1 or *Oa*PAC – in the heart induced bradycardia (**Figure 7—figure supplement 2**). A prolonged increase in intracellular $Ca^{2+}$ induced

by PKA-mediated phosphorylation of the $Ca^{2+}$ channel might contribute to the long-lasting bradycardia. Different effects of *b*PAC and *Oa*PAC activation on neurons and the heart are due to differences in basal and photo-inducible activity of these two PACs (*Ohki et al., 2016*). Further analysis is required to reveal the precise mechanism for optogenetic control of cell functions with *Be*GC1 and *b*PAC/*Oa*PAC. A two-component system comprising *b*PAC and the prokaryotic CNG potassium channel (PAC-K) silenced the activity of zebrafish neurons (*Bernal Sierra et al., 2018*). In combination with endogenous CNG or exogenous CNG $K^+$ channel, PAC can be used to both activate and inhibit zebrafish neurons.

Optogenetic control of intracellular cAMP and cGMP concentrations has been achieved in cells and tissues in which the role of cAMP and cGMP is well understood, as described previously (*Gutierrez-Triana et al., 2015*). In this study, we focused our analysis on hindbrain reticulospinal V2a neurons and cardiomyocytes, partly as a comparison with ChRs. In the future, expression of these tools in various types of cells and examination of their activity will reveal their usefulness in regulating intracellular cAMP and cGMP levels.

Cell and tissue functions are regulated by various intercellular signaling molecules, such as neurotransmitters, hormones, and cytokines. We demonstrated the usefulness of multiple types of ChRs, cGMP/cAMP-producing tools (in this study), and of G-protein-coupled rhodopsins (in the accompanying paper, *Hagio et al., 2023*) in manipulating zebrafish neuronal and cardiomyocyte function. Optogenetic studies with these tools alone or in combination will elucidate detailed mechanisms of cellular and tissue regulation.

# Materials and methods

**Key resources table**

| Reagent type (species) or resource | Designation | Source or reference | Identifiers | Additional information |
|---|---|---|---|---|
| Gene (*Guillardia theta*) | *Gt*CCR4 | *Yamauchi et al., 2017* | GenBank: MF039475.1 | |
| Gene (*Klebsormidium nitens*) | *Kn*ChR | *Tashiro et al., 2021* | GenBank: DF236986.1, GAQ79757.1 | |
| Gene (*Chlamydomonas reinhardtii*) | *Cr*ChR2[T159C] | *Berndt et al., 2011* | | |
| Gene (*Chloromonas oogama*) | *Co*ChR | *Klapoetke et al., 2014* | | |
| Gene (*Chlamydomonas noctigama*) | ChrimsonR | *Klapoetke et al., 2014* | | |
| Gene (*Blastocladiella emersonii*) | *Be*GC1 | *Scheib et al., 2015* | GenBank: KP731361.1 | |
| Gene (*G. theta*) | *Gt*ACR1 | *Govorunova et al., 2015* | GenBank: KP171708.1 | |
| Gene (*Beggiatoa* sp.) | *b*PAC | *Stierl et al., 2011* | GenBank: GU461306.2 | |
| Gene (*Oscillatoria acuminata*) | *Oa*PAC | *Ohki et al., 2016* | GenPept: WP_015149803.1 | |
| Genetic reagent (*Danio rerio*) | *mitfa*[w2/w2] | *Lister et al., 1999* | | |
| Genetic reagent (*D. rerio*) | TgBAC(vsx2:GAL4FF) | *Kimura et al., 2013* | TgBAC(vsx2:GAL4FF) nns18Tg | |
| Genetic reagent (*D. rerio*) | Tg(myl7:GAL4FF) | Accompanying paper | Tg(myl7:GAL4FF)nub38Tg | Available from M. Hibi lab |
| Genetic reagent (*D. rerio*) | Tg(UAS-hsp70l:GtCCR4-3.0-EYFP) | This paper | Tg(5xUAS-hsp70l:GtCCR4-3.0-EYFP, myl7:mCherry)nub49Tg | Available from M. Hibi lab |

*Continued on next page*

*Continued*

| Reagent type (species) or resource | Designation | Source or reference | Identifiers | Additional information |
|---|---|---|---|---|
| Genetic reagent (*D. rerio*) | *Tg(UAS:CoChR-tdTomato)* | This paper | *Tg(14xUAS-E1b:CoChR-tdTomato) nub120Tg* | Available from M. Hibi lab |
| Genetic reagent (*D. rerio*) | *Tg(UAS-hsp70l:ChrimsonR-tdTomato)* | This paper | *Tg(5xUAS-hsp70l:ChrimsonR, myl7:mCherry)nub119Tg* | Available from M. Hibi lab |
| Genetic reagent (*D. rerio*) | *Tg(UAS-hsp70l:GtCCR4-MT-P2A-TagCFP)* | This paper | *Tg(5xUAS-hsp70l:GtCCR4-MT-P2A-TagCFP, myl7:mCherry)nub50Tg* | Available from M. Hibi lab |
| Genetic reagent (*D. rerio*) | *Tg(UAS-hsp70l:KnChR-3.0-EYFP)* | This paper | *Tg(5xUAS-hsp70l:KnChR-3.0-EYFP, myl7:mCherry)nub51Tg* | Available from M. Hibi lab |
| Genetic reagent (*D. rerio*) | *Tg(UAS-hsp70l:CrChR2[T159C]-mCherry)* | This paper | *Tg(5xUAS-hsp70l:CrChR2 [T159C]-mCherry, myl7:mCherry) nub52Tg* | Available from M. Hibi lab |
| Genetic reagent (*D. rerio*) | *Tg(UAS-hsp70l:GtACR1-EYFP)* | This paper | *Tg(5xUAS-hsp70l:GtACR1-EYFP, myl7:mCherry)nub53Tg* | Available from M. Hibi lab |
| Genetic reagent (*D. rerio*) | *Tg(UAS:CoChR-tdTomato)* | This paper | *Tg(14xUAS-E1b:CoChR-tdTomato) nub120Tg* | Available from M. Hibi lab |
| Genetic reagent (*D. rerio*) | *Tg(UAS-hsp70l:ChrimsonR-tdTomato)* | This paper | *Tg(5xUAS-hsp70l:ChrimsonR-tdTomato) nub119Tg* | Available from M. Hibi Lab |
| Genetic reagent (*D. rerio*) | *Tg(UAS-hsp70l:BeGC1-EGFP)* | This paper | *Tg(5xUAS-hsp70l:BeGC1-EGFP, myl7:mCherry)nub54Tg* | Available from M. Hibi lab |
| Genetic reagent (*D. rerio*) | *Tg(UAS-hsp70l:bPAC-MT-T2A-tDimer)* | This paper | *Tg(5xUAS-hsp70l:bPAC-MT-T2A-tDimer, myl7:mCherry)nub55Tg* | Available from M. Hibi lab |
| Genetic reagent (*D. rerio*) | *Tg(UAS-hsp70l:OaPAC-Flag-P2A-TagCFP)* | This paper | *Tg(5xUAS-hsp70l: OaPAC-Flag-P2A-TagCFP, myl7:mCherry) nub56Tg* | Available from M. Hibi lab |
| Genetic reagent (*D. rerio*) | *Tg(UAS-hsp70l:GCaMP6s)* | **Muto et al., 2017** | *Tg(5xUAS-hsp70l:GCaMP6s) nkUAShspzGCaMP6s13aTg* | |
| Genetic reagent (*D. rerio*) | *Tg(elavl3:Flamindo2)* | This paper | *Tg(elavl3:Flamindo2)nub57TG* | Available from M. Hibi lab |
| Cell line (hybrid of *Rattus norvegicus* and *Mus musculus*) | ND7/23 | **Wood et al., 1990** | ECACC 92090903 | https://www.saibou.jp/en/reagents/ |
| Recombinant DNA reagent | pCS2+*Gt*CCR4-3.0-EYFP | This paper | | Available from M. Hibi lab |
| Recombinant DNA reagent | pCS2+*Gt*CCR4-MT-P2A-TagCFP | This paper | | Available from M. Hibi lab |
| Recombinant DNA reagent | pCS2+*Cr*ChR2 [T159C]-mCherry | This paper | | Available from M. Hibi lab |
| Recombinant DNA reagent | pBH-R1-R2 | Accompanying paper | | Available from M. Hibi lab |
| Recombinant DNA reagent | pENTR L1-5xUAS-hsp70l-R5 | Accompanying paper | | Available from M. Hibi lab |
| Recombinant DNA reagent | pENTR L5-*Gt*CCR4-MT-P2A-TagCFP-SV40pAS-L2 | This paper | | Available from M. Hibi lab |
| Recombinant DNA reagent | pENTR L5-*Kn*ChR-3.0-EYFP-SV40pAS -L2 | This paper | | Available from M. Hibi lab |
| Recombinant DNA reagent | pENTR L5-*Gt*ACR1-EYFP-SV40pAS-L2 | This paper | | Available from M. Hibi lab |

*Continued on next page*

*Continued*

| Reagent type (species) or resource | Designation | Source or reference | Identifiers | Additional information |
|---|---|---|---|---|
| Recombinant DNA reagent | pENTR L5-*Cr*ChR2[T159C]-mCherry-SV40pAS-L2 | This paper | | Available from M. Hibi lab |
| Recombinant DNA reagent | pENTR L5-*Be*GC1-EGFP-SV40pAS-L2 | This paper | | Available from M. Hibi lab |
| Recombinant DNA reagent | pENTR L5-*Be*GC1-EGFP-SV40pAS-L2 | This paper | | Available from M. Hibi lab |
| Recombinant DNA reagent | pENTR L5-*b*PAC-MT-T2A-tDimer-SV40pAS-L2 | This paper | | Available from M. Hibi lab |
| Recombinant DNA reagent | pENTR L5-*Oa*PAC-Flag-P2A-TagCFP-SV40pAS-L2 | This paper | | Available from M. Hibi lab |
| Antibody | Mouse monoclonal anti-Flag antibody | Sigma-Aldrich | Cat# F3165; RRID:AB_259529 | Dilution 1:500 |
| Antibody | Mouse monoclonal anti-Myc tag antibody | Santa Cruz Biotechnology | Cat# sc-40; RRID:AB_627268 | Dilution 1:500 |
| Antibody | Rat monoclonal anti-GFP antibody | Nacalai Tesque, Inc. | Cat# 04404-84; RRID:AB_10013361 | Dilution 1:500 |
| Antibody | Rabbit polyclonal anti-DsRed antibody | Takara Bio | Cat# 632496; RRID:AB_10013483 | Dilution 1:500 |
| Antibody | Goat CF488A anti-mouse IgG antibody | Biotium, Inc | Cat# 20018; RRID:AB_10557263 | Dilution 1:500 |
| Antibody | Goat CF488A anti-rat IgG antibody | Biotium, Inc | Cat# 20023; RRID: AB_10557403 | Dilution 1:500 |
| Antibody | Goat CF568 anti-rabbit IgG antibody | Biotium, Inc | Cat# 20103; RRID:AB_10558012 | Dilution 1:500 |
| Chemical compound, drug | tricaine methanesulfonate | Nacalai Tesque, Inc | Cat# 01916-32 | |
| Chemical compound, drug | low gelling temperature Type VII-A | Sigma-Aldrich | A0701 | |
| Chemical compound, drug | pentylenetetrazol | Sigma-Aldrich | Cat# P6500 | |
| Software, algorithm | SutterPatch 1.1.1 | Sutter Instrument Co. | | https://www.sutter.com/AMPLIFIERS/SutterPatch.html |
| Software, algorithm | pCLAMP10.6 | Molecular Devices | | https://support.moleculardevices.com/s/article/Axon-pCLAMP-10-Electrophysiology-Data-Acquisition-Analysis-Software-Download-Page |
| Software, algorithm | PolyScan2 | Mightex | | |
| Software, algorithm | StreamPix7 | NorPix Inc | | |
| Software, algorithm | LabVIEW | National Instruments | 2015 | https://www.ni.com/ja-jp.html |
| Software, algorithm | GraphPad Prism5 | GraphPad Software | | https://www.mdf-soft.com/ |
| Software, algorithm | VSDC Free Video Editor 6.4.7.155 | FLASH-INTEGRO LLC | | https://www.videosoftdev.com/jp |

*Continued on next page*

*Continued*

| Reagent type (species) or resource | Designation | Source or reference | Identifiers | Additional information |
|---|---|---|---|---|
| Software, algorithm | Microsoft Movies & TV | Microsoft Corp. | | https://apps.microsoft.com/store/detail/movies-tv/9WZDNCRFJ3P2 |
| Software, algorithm | QuickTime player 10.5 | Apple Inc. | | https://quicktime.softonic.jp/ |
| Software, algorithm | Fiji/ImageJ | National Institutes of Health (NIH) | | http://fiji.sc/ |
| Software, algorithm | R 3.6.1 and 4.2.1 | | | https://www.r-project.org/ |
| Software, algorithm | ggplot2 3.2.0 of R | | | https://ggplot2.tidyverse.org |
| Software, algorithm | nlme 3.1–162 of R | | | https://cran.r-project.org/web/packages/nlme/index.html |
| Software, algorithm | Bonsai | *Lopes et al., 2015* | | https://open-ephys.org/bonsai |
| Software, algorithm | Python 3.5.6 | Python Software Foundation | | https://www.python.org/ |
| Software, algorithm | Tracker Video Analysis and Modeling Tool for Physics Education 5.1.5 | | | https://physlets.org/tracker/ |
| Software, algorithm | Microsoft Excel for Mac, ver. 16.74 | Microsoft | | |
| Software, algorithm | HB_frequency.py | This paper | | Source code file |

## Cell culture

The electrophysiological assays of ChRs were performed on ND7/23cells, which are a hybrid cell line derived from neonatal rat dorsal root ganglia neurons fused with mouse neuroblastoma (*Wood et al., 1990*). ND7/23 cells were obtained from DS Pharma Biomedica, Osaka, Japan, and KAC Co. Ltd., Kyoto Japan. ND7/23 cells were grown on a collagen-coated coverslip in Dulbecco's modified Eagle's medium (Fujifilm Wako Pure Chemical Corp., Osaka, Japan) supplemented with 2.5 μM all-*trans* retinal, 5% fetal bovine serum under a 5% $CO_2$ atmosphere at 37°C. ND7/23 cells have been confirmed to be free from mycoplasma contamination, and their identity has been verified through careful morphological observation. The expression plasmids were constructed based on pCS2+ (see the 'Zebrafish' section) and were transiently transfected by using FuGENE HD (Promega, Madison, WI) according to the manufacturer's instructions. Electrophysiological recordings were then conducted 16–36 hr after transfection. Successfully transfected cells were identified by EYFP, CFP, mCherry, or tdTomato fluorescence under a microscope prior to measurements.

## Electrophysiology

All experiments were carried out at room temperature (25 ± 2°C). Photocurrents were recorded using an amplifier IPA (Sutter Instrument, Novato, CA) or Axopatch 200B amplifier (Molecular Devices, Sunnyvale, CA) under a whole-cell patch-clamp configuration. Data were filtered at 5 kHz and sampled at 10 kHz, then stored in a computer (IPA and SutterPatch, Sutter Instrument or Digdata1550 and pCLAMP10.6, Molecular Devices). The standard internal pipette solution for whole-cell voltage-clamp contained (in mM) 125 K-gluconate, 10 NaCl, 0.2 EGTA, 10 HEPES, 1 $MgCl_2$, 3 MgATP, 0.3 $Na_2GTP$, 10 $Na_2$-phosphocreatine, and 0.1 leupeptin, adjusted to pH 7.4 with KOH. The standard extracellular solution contained (in mM) 138 NaCl, 3 KCl, 10 HEPES, 4 NaOH, 2 $CaCl_2$, 1 $MgCl_2$, and 11 glucose,

adjusted to pH 7.4 with KOH. Time constants were determined by a single exponential fit unless noted otherwise.

## Optics for cultured cells

For the whole-cell patch clamp, irradiation at 385, 423, 469, 511, 555, 590, or 631 nm was carried out using Colibri7 (Carl Zeiss, Oberkochen, Germany) controlled by computer software (SutterPatch Software version 1.1.1, Sutter Instrument or pCLAMP10.6, Molecular Devices). Light power was directly measured under an objective lens of the microscope by a visible light-sensing thermopile (MIR-100Q, SSC Inc, Mie, Japan).

## Zebrafish

All Tg zebrafish lines in this study were generated using the *mitfa*$^{w2/w2}$ mutant (also known as *nacre*) line, which lacks melanophores (*Lister et al., 1999*). To construct expression plasmids for *Gt*CCR4, the cDNA of a fusion protein *Gt*CCR4-3.0-EYFP or the open-reading frame (ORF) of *Gt*CCR4 with a MycTag (MT) tag sequence, the 2A peptide sequence (P2A) from porcine teschovirus (PTV-1) (*Provost et al., 2007*; *Tanabe et al., 2010*), and TagCFP (Evrogen, Moscow, Russia) (*Gt*CCR4-MT-P2A-TagCFP) were subcloned to pCS2+. *Gt*CCR4-3.0-EYFP, which contains a membrane-trafficking signal and the ER-export signal (3.0) from the Kir2.1 potassium channel (*Gradinaru et al., 2010*; *Hoque et al., 2016*), was constructed according to a previously described procedure (*Hoque et al., 2016*). To construct *Kn*ChR expression plasmids, a carboxy terminal-truncated version (amino acids 1–272 were used), fused with the membrane- and ER-export signals and EYFP, was used (*Tashiro et al., 2021*). The *Kn*ChR-3.0-EYFP, *Cr*ChR2[T159C]-mCherry, ChrimsonR-tdTomato, *Gt*ACR1-EYFP, *Be*GC1-EGFP, *Oa*PAC, and *b*PAC-MT-T2A (2A sequence from *Thosea asigna* virus) DNA fragments were isolated by PCR from ph*Kn*ChhR-272–3.0-eYFP (*Tashiro et al., 2021*), pmCherry ChR2 T159C (*Berndt et al., 2011*), pTol1-UAS:ChrimsonR-tdTomato (*Antinucci et al., 2020*), pFUGW-h*Gt*ACR1-EYFP (*Govorunova et al., 2015*) (a gift from John Spudich [RRID:ADDgene_67795]), peGFP-N1-*Be*GC1 (*Matsubara et al., 2021*), pCold His *Oa*PAC (*Ohki et al., 2016*) (a gift from Sam-Yong Park), and pAAV hSyn1 *b*PAC cMyc T2A tDimer (a gift from Thomas Oertner [RRID:Addgene_85397]), respectively, and were subcloned to pCS2+ (for *Oa*PAC, Flag-P2A-TagCFP was attached at the carboxy-terminal). pENTR L1-R5 entry vectors containing five repeats of the upstream activation sequence (UAS) and the *hsp70l* promoter (*Muto et al., 2017*), and pENTR L5-L2 vectors containing the ORF of the optogenetic tools and the polyadenylation site of SV40 (SV40pAS) from the pCS2+ plasmids were generated by the BP reaction of the Gateway system. The UAS-hsp70l promoter and optogenetic tool expression cassettes were subcloned to the Tol2 donor vector pBleeding Heart (pBH)-R1-R2 (*Dohaku et al., 2019*), which was modified from pBH-R4-R2 and contains the mCherry cDNA and SV40 pAS under the *myosin, light chain 7, regulatory* (*myl7*) promoter (*van Ham et al., 2010*) by the LR reaction of the Gateway system. For expression of *Co*ChR-tdTomato, pTol1-UAS:*Co*ChR-tdTomato (*Antinucci et al., 2020*) was used. To make a Tol2 plasmid expressing the cAMP fluorescent indicator Flamindo2 in all postmitotic neurons, the *elavl3* promoter (*Park et al., 2000*), Flamindo2 cDNA (*Odaka et al., 2014*), and SV40pAS were subcloned to pT2ALR-Dest (pT2ALR-elavl3-Flamindo2). To make Tg fish, 25 pg of the Tol1 or Tol2 plasmids and 25 pg of Tol1 or Tol2 transposase capped and polyadenylated RNA were injected into one-cell-stage embryos. The Tg fish that expressed optogenetic tools in a Gal4-dependent manner are referred as to *Tg(UAS:opto-tool). Tg(UAS:opto-tool)* fish were crossed with *TgBAC(vsx2:GAL4FF)* (*Kimura et al., 2013*), *Tg(myl7:GAL4FF)*, or *Tg(elavl3:GAL4-VP16)* (*Kimura et al., 2013*) to express tools in the hindbrain reticulospinal V2a neurons, heart, and all postmitotic neurons, respectively. For Ca$^{2+}$ imaging and cAMP monitoring, *Tg(5xUAS-hsp70l:GCaMP6s)* (*Muto et al., 2017*) and *Tg(elavl3:Flamindo2)* were used. Adult zebrafish were raised at 28.5°C with a 14 hr light and 10 hr dark cycle. Individual larvae used for behavioral experiments were kept in the dark except for the fluorescence observation and light exposure experiments.

## Immunostaining

For immunostaining, anti-Flag antibody (1:500, mouse, Sigma-Aldrich, St. Louis, MO, Cat# F3165, RRID:AB_259529), anti-Myc Tag (MT) antibody (1:500, mouse, Santa Cruz Biotechnology, Dallas, TX, Cat# sc-40, RRID:AB_627268), anti-GFP (1:500, rat, Nacalai Tesque, Inc, Kyoto, Japan, Code: 04404-84, RRID:AB_10013361), and anti-DsRed (1:500, rabbit, Takara Bio, Shiga, Japan, Cat# 632496;

RRID:AB_10013483) were used as primary antibodies. CF488A anti-mouse IgG (1:500, H+L, Biotium Inc, Fremont, CA, Cat# 20018; RRID:AB_10557263), CF488A anti-rat IgG (1:500, H+L, Biotium, Inc, Cat# 20023; RRID:AB_10557403), and CF568 anti-rabbit IgG (1:500, H+L, Biotium Inc, Cat# 20103; RRID:AB_10558012) were used as secondary antibodies. The detailed method for immunostaining is described in the accompanying paper. Images were acquired using a confocal laser inverted microscope LSM700 (Carl Zeiss). To detect weak fluorescent signals, laser power was increased, but when the power was increased by a factor of 2 or more, it was noted in the figure legend (*Figure 7A*).

## Locomotion assay

The expression of optogenetic tools in 3-dpf larvae was determined by the expression of fluorescent marker in reticulospinal V2a neurons. Sibling fish that did not express the fluorescent marker were used as control fish. The detailed method is described in the accompanying paper. Briefly, after larvae were anesthetized with tricaine methansulfonate (Nacalai Tesque, Inc, Cat# 01916-32) and embedded in 2.5% agarose (low gelling temperature Type VII-A A0701, Sigma-Aldrich), the tail was set free by cutting the agarose around it. This agarose was placed in a 90 mm Petri dish filled with rearing water and kept for 20 min to recover from anesthesia. Light stimulation was performed using a patterned LED illuminator system LEOPARD (OPTO-LINE, Inc, Saitama, Japan) and the control software Poly-Scan2 (Mightex, Toronto, Canada) was used. The irradiation intensity and area were 0.4 mW/mm$^2$ and 0.30 mm × 0.34 mm. Tail movements were captured by an infrared CMOS camera (67 fps, GZL-CL-41C6M-C, Teledyne FLIR LLC, Wilsonville) mounted under the stage and StreamPix7 software (NorPix Inc, Montreal, Canada) and analyzed by Tracker Video Analysis and Modeling Tool for Physics Education version 5.1.5. The timing of tail motion capture and light irradiation to the reticulospinal V2a neurons was controlled by a USB DAQ device (USB-6008, National Instruments, Austin, TX) and the programming software LabVIEW (2015, National Instruments). The stimulation was repeated six times every 10 or 20 min, 100 ms (ChRs) or 500 ms (*Be*GC1 and adenylyl cyclases) each time, with a minimum of eight individuals for each strain. Trials in which swimming behavior was induced within 8 s after light stimulation were defined as induced trials. The percentage of induced trials was defined as locomotion rate, excluding trials in which swimming behavior was elicited before light stimulation. The time from the start of light irradiation to the first tail movement was defined as latency, and the time from the start of the first tail movement to the end of that movement was defined as duration. The maximum distance the tail moved from the midline divided by body length was defined as strength. To examine the tools' ability to inhibit locomotion, 4-dpf Tg larvae were pretreated with 15 mM penty-lenetetrazol (Sigma-Aldrich, Cat# P6500) and spontaneous tail movements were induced by white LED light (peak 640 nm; Kingbright Electronic Co., Ltd., New Taipei City, Taiwan) powered by a DC power supply (E3631A; Agilent Technologies, Santa Clara, CA) for 5 s. After 500 ms from the onset of the white LED light, the hindbrain reticulospinal V2a neurons were stimulated with the patterned LED illuminator. Trials in which swimming behavior stopped within 1 s after white light stimulation were defined as locomotion-inhibition trials. The percentage of locomotion-inhibition trials was calculated; these values are indicated in *Table 1*. Graphs were created with GraphPad Prism5 software (GraphPad Software, San Diego, CA). We used VSDC Free Video Editor version 6.4.7.155 (FLASH-INTEGRO LLC, Moscow, Russia) and Microsoft Movies & TV (Microsoft Corp., Redmond, WA) to make all movies.

## Heartbeat experiments

4-dpf larvae carrying an expressed fluorescent marker in the heart were used for the experiments. Sibling fish that did not express the marker were used as control fish. Four larvae were used for each line. The detailed method is described in the accompanying paper. Briefly, after larvae were quickly anesthetized with about 0.2% tricaine methanesulfonate and embedded in agarose, they were placed in a 90 mm Petri dish filled with water and kept for 20 min to recover from anesthesia. Irradiation intensity was adjusted to 0.5 mW/mm$^2$. The area of irradiation was 0.17 mm × 0.25 mm, including the entire heart. The heart area in Tg fish was irradiated for 100 ms (ChRs) or 5 s (*b*PAC-MT) with light wavelengths that had the closest values to the maximum absorption wavelength of each optogenetic tool, as shown in *Table 1*. The HBs of larvae were captured by an infrared CMOS camera (67 fps) and recorded with StreamPix7 software, as described above. The irradiation trial was repeated six times every 3 min (for *Gt*CCR4-3.0-EYFP and *Gt*ACR1-EYFP) or 10 min (for *Kn*ChR-3.0-EYFP, *Cr*ChR2[T159C]-mCherry, *Co*ChR-tdTomato, ChrimsonR-tdTomato,

and *b*PAC-MT) for one fish and a total of four larvae were analyzed for each strain. The video recordings of the HBs were observed using QuickTime player 10.5 (Apple Inc, Cupertino, CA). After opening videos with Fiji/ImageJ (National Institutes of Health, Bethesda, MD), the entire heart was set as the region of interest (ROI), and the luminosity (AU: arbitrary units) data in the ROI was used to create graphs of HBs using ggplot2 version 3.2.0 in R. To calculate relative HB frequency, temporal changes in luminosity were obtained from the video using Bosai (*Lopes et al., 2015*) and the frames where HBs occurred were identified by the code (HB_frequency.py) created in Python version 3.5.6 (Python Software Foundation, Wilmington, DE). Relative HB frequency was calculated from the HB frame data, 500 ms before and after each time point using Excel (Microsoft). Graphs of the average relative HB frequency were created by ggplot2 of R. The latency to cardiac arrest and the time to first resumption of HBs were also measured. Graphs were created with GraphPad Prism5 software. All movies were created with VSDC Free Video Editor. Simple HB experiments were also performed using a light source equipped with an MZ16 FA microscope and GFP (460–500 nm), YFP (490–510 nm), and DSR filters (530–560 nm, Leica, Wetzlar, Germany), as indicated in *Table 1*.

## Ca²⁺ imaging

4-dpf Tg fish expressing *Kn*ChR or *Gt*ACR1, and GCaMP6s in cardiomyocytes were used. Tg fish expressing only GCaMP6s were used as controls. The larvae anesthetized with tricaine methanesulfonate were embedded in 3% agarose (low-gelling temperature Type VII-A, Sigma-Aldrich) in 1/10 Evans solution, placed in a 90 mm Petri dish filled with water, and left on the microscope stage for 10 min. A 130 W light source (U-HGLGPS, Olympus, Tokyo, Japan) with a fluorescence detection filter (excitation 470–495 nm, emission 510–550 nm, U-MNIBA3, Olympus) was used to observe the fluorescence of GCaMP6s. A CCD camera (ORCA-R2, Hamamatsu Photonics, Shizuoka, Japan) located on the microscope was used to capture the GCaMP6s fluorescence images at 9 fps. After image acquisition, the entire heart area was manually set as the ROI using Fiji/ImageJ, and fluorescence intensity was measured. Trials were repeated three times every 10 min. The relative change in ΔF/F was calculated by dividing the fluorescence intensity in each frame by the fluorescence intensity at the start of light exposure.

## cAMP live imaging

3-dpf larvae expressing an optogenetic tool and the cAMP indicator Flamindo2 in postmitotic neurons were used. Sibling larvae that did not express the optogenetic tool were used as controls. The larvae that were quickly anesthetized with 0.04% tricaine methanesulfonate were embedded in 3% agarose, placed in a 90 mm Petri dish filled with rearing water, and left on the microscope stage for 20 min. A 130 W light source (U-HGLGPS, Olympus) with a fluorescence detection filter (excitation 470–495 nm, emission 510–550 nm) was used for observation. The fluorescence images were captured by a CCD camera (ORCA-R2, Hamamatsu Photonics) at 9 fps. After image acquisition, the entire optic tectum area was set as a ROI using Fiji/ImageJ and fluorescence intensity was measured. ΔF/F was calculated.

## Statistical analysis

Data were analyzed using R software package (versions 3.6.1 and 4.2.1). Statistical tests were applied as indicated in figure legends. All data in the text and figures are expressed as the mean ± standard error of the mean (SEM). A linear mixed-effects model was applied using R package 'nlme' version 1.3–162.

## Acknowledgements

We thank Shin-ichi Higashijima, Koichi Kawakami, and the National Bioresource Project for providing transgenic fish; John Spudich, Sam-Yong Park, Thomas Oertner, and Isaac Bianco for providing plasmid DNAs; Tamiko Itoh for managing fish mating and care; Ryosuke Takeuchi for helping us analyze heartbeat experiments. We also thank the members of the Kandori and Hibi laboratories for helpful discussion. MEXT KAKENHI JP26115512, JSPS KAKENHI JP18H02448 (to MH), JP18K06333 (to TS), CREST Japan Science and Technology Agency (JST) JPMJCR1753 (to HK and MH).

## Additional information

### Funding

| Funder | Grant reference number | Author |
|---|---|---|
| Japan Science and Technology Agency | JPMJCR1753 | Masahiko Hibi Hideki Kandori |
| Japan Society for the Promotion of Science | JP26115512 | Masahiko Hibi |
| Japan Society for the Promotion of Science | JP18K06333 | Takashi Shimizu |
| Japan Society for the Promotion of Science | JP18H02448 | Masahiko Hibi Hideki Kandori |

The funders had no role in study design, data collection and interpretation, or the decision to submit the work for publication.

### Author contributions

Hanako Hagio, Shoko Hososhima, Data curation, Formal analysis, Investigation, Methodology, Writing – review and editing; Wataru Koyama, Aysenur Deniz Song, Data curation, Formal analysis, Investigation; Shiori Hosaka, Data curation, Formal analysis, Investigation, Methodology; Janchiv Narantsatsral, Koji Matsuda, Formal analysis, Investigation; Takashi Shimizu, Methodology, Supervision, Writing – review and editing; Satoshi P Tsunoda, Validation, Writing – review and editing; Hideki Kandori, Conceptualization, Validation, Funding acquisition, Writing – review and editing; Masahiko Hibi, Conceptualization, Data curation, Methodology, Funding acquisition, Writing - original draft, Writing – review and editing

### Author ORCIDs

Hanako Hagio (ID) http://orcid.org/0000-0003-2197-4595
Wataru Koyama (ID) http://orcid.org/0009-0005-6851-5961
Aysenur Deniz Song (ID) http://orcid.org/0000-0003-0998-9225
Takashi Shimizu (ID) http://orcid.org/0000-0002-8750-6797
Hideki Kandori (ID) http://orcid.org/0000-0002-4922-1344
Masahiko Hibi (ID) http://orcid.org/0000-0002-9142-4444

### Ethics

The animal experiments in this study were approved by the Nagoya University Animal Experiment Committee and were conducted in accordance with the Regulation on Animal Experiments from Nagoya University.

### Decision letter and Author response

Decision letter https://doi.org/10.7554/eLife.83975.sa1
Author response https://doi.org/10.7554/eLife.83975.sa2

## Additional files

### Supplementary files

• MDAR checklist

• Source code 1. Software that detects the timing of each heartbeat from the data of luminosity changes over time.

### Data availability

All data generated or analysed during this study are included in the manuscript and supporting files and source data files have been provided for Figures 1-7.

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
