## [Editor Report]

This manuscript provides a valuable resource for scientists who wish to manipulate second messengers in zebrafish using optogenetics. The authors provide solid evidence, based on behaviour, monitoring of heartbeat and imaging, that several of the tools tested can have an effect in larval fish. Tools that lack an effect are also described. As the tools affect second messengers that are used in multiple cell types, the results should be of interest to scientists working in a variety of areas.

---

## [Decision Letter]

**Decision letter after peer review:**

Thank you for submitting your article "Optogenetic manipulation of second messengers in neurons and cardiomyocytes with microbial rhodopsins and adenylyl cyclase" for consideration by *eLife*. Your article has been reviewed by 3 peer reviewers, one of whom is a member of our Board of Reviewing Editors, and the evaluation has been overseen by Didier Stainier as the Senior Editor. The reviewers have opted to remain anonymous.

Essential revisions:

1) Provide an analysis of the effects of different light intensities, on both behaviour and neuronal activity.

2) Provide a comparison with existing optogenetic tools, e.g. *Co*ChR.

3) Provide a comparison with opsin-negative animals. The long latency gives rise to the possibility that some of the responses are not due to the transgene.

4) Rewrite the manuscript, especially the introduction, to more accurately reflect the scope of the work. A focus on second messengers necessitates analysis of the second messengers.

*Reviewer #1 (Recommendations for the authors):*

In introducing bPac and OaPac, the authors note that both tools have been tested in a variety of species and were found to be useful. The suggestion that the tools warrant further testing because their activity in other cell types and species are unknown (last line, first paragraph on page 7), is illogical. The unstated assumption here is that zebrafish provides the definitive test for all cell types and all vertebrates. The data provide strong evidence that these tools work in two cell types in the zebrafish, but the relevance to other species cannot be extrapolated based on the authors' own reasoning. It is suggested that the rationale for the experiments be rephrased.

The limitations of KnChR, in terms of not being selective to a single ion, points to what seems to be a weakness. Specifically, as currently written, the manuscript does not appear to have a strong central question/idea, but does contain a number of interesting findings related to optogenetic tools that work in zebrafish. It is strongly recommended that framework of the manuscript be recrafted.

The manuscript would be strengthened with a consideration of whether the tools tested here are more useful than tools that are currently in use, or where the tool provides some new insight. While a comparison has been done for neuronal depolarization with KnChR, this experiment is not relevant to testing the role of specific second messengers. For the case of Pac, it should be noted that bPac has been tested previously in the zebrafish nervous system, in the context of the PAC-K system (https://www.nature.com/articles/s41467-018-07038-8). It would also be informative to readers who are not specialists in the cardiac system to know how changing cGMP levels would be expected to affect heart beat.

A strength of the paper is the use of indicators such as GCaMP6 and flamindo2. It would be informative to monitor cGMP directly e.g. with Green cGull (https://pubmed.ncbi.nlm.nih.gov/28722423/) or the more recent red version.

*Reviewer #2 (Recommendations for the authors):*

1. As a method for assessing expression levels, immunohistochemistry is only semi-quantitative. It is not mentioned how detectable the expression is under the epifluorescent stereomicroscope in vivo. The authors should comment on that. Also, information regarding the expression when using other driver lines and the level of mosaicism would be very helpful, if included in the manuscript.

2. It is great to see such big effects on behavior. However, it remains unclear to what extent the target cells have directly been activated. In other studies on the application of optogenetics, measurements of neuronal activity changes and their light dependency are oftentimes included (e.g. see Antinucci et al. *eLife* 2020). How reliably can spiking be evoked? Such measurements would be very informative and raise the strength of the evidence from "solid" to "convincing/compelling".

3. Inclusion of more visual illustrations of the experimental setup and the mode of action of each optogenetic protein in the Figures would help guide the reader through the manuscript. Not every reader will be familiar with the names of the presented proteins or their way of action.

4. The adenylyl and guanylyl cyclase optogenetic tools are presented and discussed as possible valuable manipulators of neuronal excitability. However, it seems to the reviewer, that such tools are best used when the nucleotide second messengers are indeed the target signal in the experiment (e.g. the cited Gutierrez-Triana et al. 2015). For the fast control of membrane potential, the available optogenetic channels and pumps should be ideal. I suggest revising the wording in the manuscript accordingly.

*Reviewer #3 (Recommendations for the authors):*

Some specific points:

The latency (and bimodal distribution) of the behavioral responses raises serious doubts and suggests to me that at least a substantial fraction of responses are not a result of direct optogenetic activation of V2a neurons. As the authors have collected data for opsin negative control animals, they should analyze the latency distribution with a view to seeing if opsin-independent swims arise at longer latencies (e.g. >100 ms). This could guide selection of a latency threshold that could be applied to the data in Figure 2.

The long latency of swims evoked in the BeGC, bPAC and OaPAC experiments is also a concern. Here, all swims appear to be long latency with none <100 ms. Although cAMP/cGMP modulation might be expected to drive behavior at longer latencies that channelrhodopsins, it is unclear how changes in cAMP/cGMP leads to V2a firing, especially with mean latency exceeding 2 s. Although the authors speculate about a CNG-mediated mechanism, no data is presented. Electrophysiological recordings and/or pharmacology would be informative here.

The paper is motivated by the need to "precisely control second messengers in vivo", and the introduction and discussion talk at length about the different ion selectivity of channelrhodopsins. I was surprised by how little assessment there was of either of these factors in V2a cells or cardiomyocytes. With so little assessment of second messenger pathways, I consider this claim goes beyond what the manuscript has accomplished: "We demonstrated the usefulness of multiple types of channelrhodopsins, cGMP/cAMP-producing tools (in this study), and of GPCR rhodopsins (in the accompanying paper) to manipulate second messenger signaling in zebrafish neurons and cardiomyocytes." I'd suggest rewriting the manuscript with a more accurate focus on the scope of the work presented.

In figure 1, the sensitivity (photocurrent vs light intensity) analysis is unclear. What are units of EC50? Why is no response curve shown?

Photocurrent amplitude data is not convincing. Why is N so low? For GtCCR4-YFP two datapoints are far greater than the remaining four. This raises suspicions and calls for more data. The current data is not suitably represented by an arithmetic mean and (what I presume are) SEM error bars.

In assessing rates of optogenetically evoked responses, data are presented without error bars (i.e. Figure 2C, Figure 5C, Fig6C). This suggests trials have been pooled across animals but error should be computed across the biological replicates in each group and the statistical analysis should be similarly revised.

F3A: There seems to be a substantial difference in heart size between these examples. Is the scale the same? If so, what accounts for this?

In several places the description of results needs to made more precise and specific.

– In text describing figure 1: `suggesting that high-frequency photostimulation is possible for these channelrhodopsins`. What is meant by "high"? Its a rather intermediate tau-OFF compared to other opsins, and much slower than fast opsins such as Chronos.

– "Optical activation of GtCCR4 and KnChR in the hindbrain reticulospinal V2a neurons, which are involved in locomotion, immediately induced swimming behavior": This statement is misleading both because responses are can never be "immediate" and because the latency of e.g. GtCCR4 responses are often very slow. Similarly, "Light stimulation of the reticulospinal V2a neurons with CrChR2[T159C]-mCherry immediately evoked tail movements": Please be more precise and give the mean latency, in milliseconds.

– In the discussion: "activation with BeGC1 and PACs induced neural activation with a short delay". This seems a strange statement as V2a-evoked behavioral responses had very long latency.

---

## [Author Response]

Essential revisions:1) Provide an analysis of the effects of different light intensities, on both behaviour and neuronal activity.

We stimulated neuronal ND7/23 cells, reticulospinal V2a neurons or cardiomyocytes expressing microbial optogenetic tools at various light intensities and examined their effects on neuronal activities and behaviors (tail movements and cardiac arrest). These data are shown in revised Figure 1, Figure 1-supplement 1, Figure 3, Figure 3-supplements 2, 3, Figure 5, and Figure 5-supplements 1, 2. We described the data on page 12, line 20-page 13, line 1 and page 14, lines 10-13 in the revised manuscript.

2) Provide a comparison with existing optogenetic tools, e.g. *Co*ChR.

We examined the activity of *Co*ChR and ChrimsonR in neuronal ND7/23 cells. In addition, we generated transgenic zebrafish expressing *Co*ChR or ChrimsonR, and examined their activity in V2a neurons and cardiomyocytes. We thereby compared the activity of *Gt*ACR4, *Kn*ChR, and *Cr*ChR2[T159C] with that of *Co*ChR and ChrimsonR. The data are shown in Figure 1, Figure 1-supplement 1, Figure 2, Figure 3, Figure 3-supplement 3, and Figure 5-supplement 2. We described the data for *Co*ChR and ChrimsonR in the relevant part of the Result section (pages 8-14) and discussed a comparison on page 18, lines 3-16 in the revised manuscript.

We found that *Kn*ChR was a more potent optogenetic tool than *Gt*CCR4, *Cr*ChR2, and ChrimsonR in zebrafish reticulospinal V2a neurons. Optogenetic activity of *Kn*ChR was comparable to that of *Co*ChR in both reticulospinal V2a neurons and cardiomyocytes (Figures 1, 3, 5). Truncation of *Kn*ChR prolonged the channel open lifetime by more than 10-fold (*Tashiro et al. , 2021*) (Figure 1). *Kn*ChR conducts various monovalent and bivalent cations, including H^+^, Na^+^, and ca^2+^, while *Kn*ChR has a higher permeability to Na^+^ and ca^2+^ and a higher permeability ratio of ca^2+^ to Na^+^ than *Cr*ChR2 (*Tashiro et al. , 2021*). These properties may contribute to the high photo-inducible activity of *Kn*ChR. Activation of *Kn*ChR may induce influx of more cations with a longer channel open time than *Cr*ChR2 and ChrimsonR, leading to stronger cell depolarization. Optogenetic activity of *Kn*ChR was comparable to that of *Gt*CCR4 in cultured cells, but higher than *Gt*CCR4 in zebrafish reticulospinal V2a neurons and cardiomyocytes. While the exact reason is unclear, it is possible that the expression of functional *Kn*ChR protein may be high in zebrafish cells.

3) Provide a comparison with opsin-negative animals. The long latency gives rise to the possibility that some of the responses are not due to the transgene.

We compared the latency of zebrafish larvae expressing each tool with those larvae not expressing the tool. The data are shown in Figure 3, Figure 3-supplement 1, Figure 5, Figure 6, Figure 7, and Figure 7-supplement 1. Statistically, we considered responses within 8 s after the start of light stimulation as positive, and significant differences in responses were observed depending on the presence or absence of tool expression, suggesting that tail movements were induced by tool activation. In the absence of tool expression, spontaneous movements were occasionally observed, but they did not often occur within 8 s. We described the data on page 15, line 20-page16, line 4 in the revised manuscript.

4) Rewrite the manuscript, especially the introduction, to more accurately reflect the scope of the work. A focus on second messengers necessitates analysis of the second messengers.

We agree with the reviewers and the editor. We changed the title to “Optogenetic manipulation of neuronal and cardiomyocyte functions in zebrafish using microbial rhodopsins and adenylyl cyclases” and revised the abstract and introduction accordingly. We explained in the introduction section that the purpose of the study is to understand cell and tissue function through the optical control of intracellular ions and cAMP/cGMP and to examine their effects.

Reviewer #1 (Recommendations for the authors):In introducing bPac and OaPac, the authors note that both tools have been tested in a variety of species and were found to be useful. The suggestion that the tools warrant further testing because their activity in other cell types and species are unknown (last line, first paragraph on page 7), is illogical. The unstated assumption here is that zebrafish provides the definitive test for all cell types and all vertebrates. The data provide strong evidence that these tools work in two cell types in the zebrafish, but the relevance to other species cannot be extrapolated based on the authors' own reasoning. It is suggested that the rationale for the experiments be rephrased.

We have added a sentence “Specifically, the effectiveness of *b*PAC and *Oa*PAC in a variety of zebrafish cells remains unclear” to the end of this paragraph, to explain the purpose of this study.

The limitations of KnChR, in terms of not being selective to a single ion, points to what seems to be a weakness. Specifically, as currently written, the manuscript does not appear to have a strong central question/idea, but does contain a number of interesting findings related to optogenetic tools that work in zebrafish. It is strongly recommended that framework of the manuscript be recrafted.

Instead of making “second messenger regulation” the main focus of the manuscript, we have restructured the entire manuscript to focus on the optogenetic control of zebrafish neurons and cardiomyocytes.

The manuscript would be strengthened with a consideration of whether the tools tested here are more useful than tools that are currently in use, or where the tool provides some new insight. While a comparison has been done for neuronal depolarization with KnChR, this experiment is not relevant to testing the role of specific second messengers. For the case of Pac, it should be noted that bPac has been tested previously in the zebrafish nervous system, in the context of the PAC-K system (https://www.nature.com/articles/s41467-018-07038-8). It would also be informative to readers who are not specialists in the cardiac system to know how changing cGMP levels would be expected to affect heart beat.

We have now cited the paper by Bernal Sierra et al., 2018 describing a two-component optical silencer system comprising PACs and the small cyclic nucleotide-gated potassium channel SthK, (e.g. fused bPAC-K), and discussed the use of PAC to activate and inhibit zebrafish neurons on page 21, lines 3-7.

A strength of the paper is the use of indicators such as GCaMP6 and flamindo2. It would be informative to monitor cGMP directly e.g. with Green cGull (https://pubmed.ncbi.nlm.nih.gov/28722423/) or the more recent red version.

In addition to flamindo2, we attempted to express cAMP indicator R-FlincA (Ohta et al., Sci Rep *8*(1), 1866, 2018) and cGull (Matsuda et al., ACS Sens 2(1):46-52, 2017). We attempted to transiently express these in early-stage zebrafish embryos using the CMV promoter, but no expression was observed. We also attempted to generate transgenic fish expressing these indicators using the *elavl3* promoter, but failed to establish Tg fish that expressed them. It is plausible that these indicators are not stable in zebrafish cells. Since these are negative data, we decided not to describe them in this paper.

Reviewer #2 (Recommendations for the authors):1. As a method for assessing expression levels, immunohistochemistry is only semi-quantitative. It is not mentioned how detectable the expression is under the epifluorescent stereomicroscope in vivo. The authors should comment on that. Also, information regarding the expression when using other driver lines and the level of mosaicism would be very helpful, if included in the manuscript.

We were able to detect the expression of the tools (excluding *b*PAC and *Oa*PAC, which were not fused with fluorescent proteins) in the hindbrain reticulospinal V2a neurons and cardiomyocytes of all living transgenic zebrafish in vivo under an epifluorescent stereomicroscope. We described this on page 10, lines 19-21, and page 13, lines 13-15.

We quantified the expression levels by immunostaining with anti-fluorescent marker antibodies or anti-Tag antibodies (for *b*PAC and *Oa*PAC, *Gt*CCR4-MT), which we have detailed in Table 1. Notably, among the transgenic lines, *Kn*ChR was the only one that displayed clear mosaic expression. We described this on page 11, lines 4-5.

2. It is great to see such big effects on behavior. However, it remains unclear to what extent the target cells have directly been activated. In other studies on the application of optogenetics, measurements of neuronal activity changes and their light dependency are oftentimes included (e.g. see Antinucci et al. eLife 2020). How reliably can spiking be evoked? Such measurements would be very informative and raise the strength of the evidence from "solid" to "convincing/compelling".

We agree with the reviewer that electrophysical measurements are very informative. However, because we are unfamiliar with the electrophysiological analysis of zebrafish cells, and because it would take a great deal of time to produce accurate data in many transgenic fish, we decided not to conduct electrophysiology in vivo assessments in this study.

3. Inclusion of more visual illustrations of the experimental setup and the mode of action of each optogenetic protein in the Figures would help guide the reader through the manuscript. Not every reader will be familiar with the names of the presented proteins or their way of action.

We have prepared a diagram of the experimental setup and the optogenetic tools used in the study, which is included as Figure 2A and 2B.

4. The adenylyl and guanylyl cyclase optogenetic tools are presented and discussed as possible valuable manipulators of neuronal excitability. However, it seems to the reviewer, that such tools are best used when the nucleotide second messengers are indeed the target signal in the experiment (e.g. the cited Gutierrez-Triana et al. 2015). For the fast control of membrane potential, the available optogenetic channels and pumps should be ideal. I suggest revising the wording in the manuscript accordingly.

For testing tools that regulate intracellular cAMP and cGMP concentrations, it may be better to test them in cells and tissues where the role of cAMP and cGMP is well understood, as described in Gutierrez-Triana et al. 2015. However, we focused our analysis on reticulospinal V2a neurons and cardiomyocytes, partly as a comparison with channelrhodopsins. We discussed this on page 21, lines 8-14 in the Discussion section.

Reviewer #3 (Recommendations for the authors):Some specific points:The latency (and bimodal distribution) of the behavioral responses raises serious doubts and suggests to me that at least a substantial fraction of responses are not a result of direct optogenetic activation of V2a neurons. As the authors have collected data for opsin negative control animals, they should analyze the latency distribution with a view to seeing if opsin-independent swims arise at longer latencies (e.g. >100 ms). This could guide selection of a latency threshold that could be applied to the data in Figure 2.

We compared the latency of zebrafish larvae expressing each tool with those larvae not expressing the tool. The data are shown in Figure 3, Figure 3-supplement 3, Figure 5, Figure 6, Figure 7, and Figure 7-supplement 1. Statistically, we considered responses within 8 s after the start of light stimulation as positive, and significant differences in responses were observed depending on the presence or absence of tool expression, suggesting that tail movements were induced by tool activation. In the absence of tool expression, spontaneous movements were occasionally observed, but they did not often occur within 8 s. We described the data on page 15, line 20-page 16, line 4 in the revised manuscript.

The long latency of swims evoked in the BeGC, bPAC and OaPAC experiments is also a concern. Here, all swims appear to be long latency with none <100 ms. Although cAMP/cGMP modulation might be expected to drive behavior at longer latencies that channelrhodopsins, it is unclear how changes in cAMP/cGMP leads to V2a firing, especially with mean latency exceeding 2 s. Although the authors speculate about a CNG-mediated mechanism, no data is presented. Electrophysiological recordings and/or pharmacology would be informative here.

We agree with the reviewer that electrophysiological experiments would be necessary to fully understand the mechanisms by which photostimulation of *Be*GC, *b*PAC and *Oa*PAC leads to neuronal activation. However, because we are unfamiliar with the electrophysiological analysis of zebrafish cells and because it would take a great deal of time to produce accurate data in many transgenic fish, we decided not to conduct electrophysiology in vivo assessments in this study.

The paper is motivated by the need to "precisely control second messengers in vivo", and the introduction and discussion talk at length about the different ion selectivity of channelrhodopsins. I was surprised by how little assessment there was of either of these factors in V2a cells or cardiomyocytes. With so little assessment of second messenger pathways, I consider this claim goes beyond what the manuscript has accomplished: "We demonstrated the usefulness of multiple types of channelrhodopsins, cGMP/cAMP-producing tools (in this study), and of GPCR rhodopsins (in the accompanying paper) to manipulate second messenger signaling in zebrafish neurons and cardiomyocytes." I'd suggest rewriting the manuscript with a more accurate focus on the scope of the work presented.

Instead of making “second messenger regulation” the main focus of the manuscript, we have restructured the entire manuscript to focus on the optogenetic control of zebrafish neurons and cardiomyocytes.

In figure 1, the sensitivity (photocurrent vs light intensity) analysis is unclear. What are units of EC50? Why is no response curve shown?

The units of EC_50_ in Figure 1E are mW/mm^2^. We have shown the response curve in Figure 1-supplement 1.

Photocurrent amplitude data is not convincing. Why is N so low? For GtCCR4-YFP two datapoints are far greater than the remaining four. This raises suspicions and calls for more data. The current data is not suitably represented by an arithmetic mean and (what I presume are) SEM error bars.

We have collected more data on the photocurrent amplitudes for each tool and revised Figure 1B. Error bars in Figure 1 indicate SEM.

In assessing rates of optogenetically evoked responses, data are presented without error bars (i.e. Figure 2C, Figure 5C, Fig6C). This suggests trials have been pooled across animals but error should be computed across the biological replicates in each group and the statistical analysis should be similarly revised.

In our revision, we have shown the rate of optogenetically evoked responses (locomotion or cardiac arrest rate) per individual animal in the revised Figure 3, Figure 3-supplements 2, 3, Figure 5, Figure 5-supplements 1, 2, Figure 6, and Figure 7.

F3A: There seems to be a substantial difference in heart size between these examples. Is the scale the same? If so, what accounts for this?

The magnification of the heart image in Figure 4A (original Figure 3A) is the same. There are individual differences in heart size.

In several places the description of results needs to made more precise and specific.– In text describing figure 1: `suggesting that high-frequency photostimulation is possible for these channelrhodopsins`. What is meant by "high"? Its a rather intermediate tau-OFF compared to other opsins, and much slower than fast opsins such as Chronos.

The low Tau-OFF values indicate faster recovery after photostimulation, which allows for early re-stimulation after recovery and therefore a higher frequency of photostimulation. However, in the Results section, we have described the data without any interpretation.

– "Optical activation of GtCCR4 and KnChR in the hindbrain reticulospinal V2a neurons, which are involved in locomotion, immediately induced swimming behavior": This statement is misleading both because responses are can never be "immediate" and because the latency of e.g. GtCCR4 responses are often very slow. Similarly, "Light stimulation of the reticulospinal V2a neurons with CrChR2[T159C]-mCherry immediately evoked tail movements": Please be more precise and give the mean latency, in milliseconds.

Instead of using the word "immediate," we explained the optogenetically evoked locomotion responses by providing latency data in the Results section as follows. “Light stimulation with *Gt*CCR4-3.0-EYFP and *Gt*CCR4-MT evoked tail movements at comparable locomotion rates, although it took more time than *Cr*ChR2[T159C]-mCherry (*Gt*CCR4-3.0-EYFP locomotion rate 62.5 ± 8.77%, latency 1.59 ± 0.536 s; *Gt*CCR4-MT locomotion rate 50 ± 8.91%, latency 1.16 ± 0.445 s, Figure 3A, B, Figure 3-supplement 1, Figure 3-videos 2, 3)”.

– In the discussion: "activation with BeGC1 and PACs induced neural activation with a short delay". This seems a strange statement as V2a-evoked behavioral responses had very long latency.

As noted above, instead of using the ambiguous term “short”, we explained the data by providing latency data in the Results section. In the Discussion, we changed the sentence to “activation with *Be*GC1 and PACs induced neural activation with a relatively long delay”.